# De-etiolation-induced protein 1 (DEIP1) mediates assembly of the cytochrome $b_6f$ complex in Arabidopsis

Omar Sandoval-Ibáñez [1], David Rolo [1], Rabea Ghandour [1], Alexander P. Hertle [1], Tegan Armarego-Marriott[1], Arun Sampathkumar [1], Reimo Zoschke [1] & Ralph Bock [1✉]

The conversion of light energy to chemical energy by photosynthesis requires the concerted action of large protein complexes in the thylakoid membrane. Recent work has provided fundamental insights into the three-dimensional structure of these complexes, but how they are assembled from hundreds of parts remains poorly understood. Particularly little is known about the biogenesis of the cytochrome $b_6f$ complex (Cyt$b_6f$), the redox-coupling complex that interconnects the two photosystems. Here we report the identification of a factor that guides the assembly of Cyt$b_6f$ in thylakoids of chloroplasts. The protein, DE-ETIOLATION-INDUCED PROTEIN 1 (DEIP1), resides in the thylakoid membrane and is essential for photoautotrophic growth. Knock-out mutants show a specific loss of Cyt$b_6f$, and are defective in complex assembly. We demonstrate that DEIP1 interacts with the two cytochrome subunits of the complex, PetA and PetB, and mediates the assembly of intermediates in Cyt$b_6f$ biogenesis. The identification of DEIP1 provides an entry point into the study of the assembly pathway of a crucial complex in photosynthetic electron transfer.

[1] Max Planck Institute of Molecular Plant Physiology, Am Mühlenberg 1, 14476 Potsdam-Golm, Germany. ✉email: rbock@mpimp-golm.mpg.de

The thylakoidal protein complexes involved in photo-synthetic electron transfer represent large macromolecular assemblies comprised of plastid genome-encoded and nuclear genome-encoded protein subunits. In addition, they bind a large number of pigments (chlorophylls, carotenoids), lipids, and redox-active co-factors such as hemes and iron-sulfur clusters[1–3]. Recent work has provided high-resolution three-dimensional structures of the complexes. The biogenesis of the complexes and their assembly from hundreds of individual parts represents a challenging task involving (i) the co-ordinated expression of nucleus-encoded and plastid-encoded protein sub-units, (ii) the post-translational import of the nucleus-encoded subunits into the chloroplast and their faithful targeting to the thylakoid membrane, (iii) the sequential integration of the protein subunits into the membrane in a spatially and temporally highly controlled manner, and (iv) the synthesis and co-ordinated insertion of a large number and variety of co-factors into the nascent complexes[4–6]. Little is known about the molecular mechanisms involved, but it has become clear that complex biogenesis depends on numerous accessory proteinaceous factors that collectively are referred to as assembly factors. They act as chaperones that support protein folding, promote protein-protein interactions and/or subunit insertion into membranes, help with co-factor attachment to protein subunits, or protect assembly intermediates from degradation or photo-oxidative damage[7,8].

Several assembly factors have been identified for photosystem II (PSII; e.g., SCO2, HCF247, HCF136, and PAM68[9–13]), photosystem I (PSI; e.g., Ycf3, Y3IP1, PYG7, and Ycf4[14–18]), and the chloroplast ATP synthase (e.g., BFA3 and CGL160[19,20]). Based on mutant analyses, protein interaction studies and identification of assembly intermediates, assembly pathways for these complexes have been proposed[21–23]. By contrast, little is known about the assembly of the cytochrome $b_6f$ (Cyt$b_6f$) complex. Cyt$b_6f$ is a dimer comprised of two identical monomeric complexes. Each monomer consists of a core formed by the chloroplast-encoded proteins PetA (cytochrome f), PetB (cytochrome $b_6$) and PetD (subunit IV), and the nucleus-encoded subunit PETC (Rieske iron-sulfur protein)[24,25]. Additional small polypeptide subunits are localized at the periphery of the complex and are believed to contribute to complex stabilization and/or regulate complex activity: the plastid-encoded small subunits PetG, PetL, and PetN, and the nucleus-encoded PETM[26–29].

In *Chlamydomonas*, several proteins involved in biogenesis of co-factors of Cyt$b_6f$ have been described, including the CCB pathway (Cofactor assembly, complex C ($b_6f$), subunit B)[30], composed of the CCB1- 4 proteins that deliver heme to the cytochrome $b_6$ apoprotein[31]. Additional proteins such as CPLD38 and CPLD49 have been implicated in Cyt$b_6f$ accumulation by a yet undefined mechanism[32,33].

Homologs of the CCB pathway genes encoding plastid-localized proteins have been identified also in the vascular plant *Arabidopsis thaliana*, and the corresponding knock-out mutants display severe growth phenotypes[34,35]. Besides the CCB homologs, only a handful of genes have been described to influence the content of the Cyt$b_6f$ complex in plants. Loss-of-function mutants for the locus *DEFECTIVE ACCUMULATION OF Cyt$b_6f$* (*DAC*) display decreased accumulation of the Cyt$b_6f$ complex (to 10–20% of wild-type levels) and exhibit pale phenotypes[36]. DAC appears to interact with PetD, but its mode of action and its possible contribution to Cyt$b_6f$ assembly or stability has remained unknown. The DnaJ-like protein *HIGH CHLOROPHYLL FLUORESCENCE 222* (HCF222) and the thioredoxin-like protein *HIGH CHLOROPHYLL FLUORESCENCE 164* (HCF164) were also shown to influence Cyt$b_6f$ accumulation, likely by their chaperone and disulfide reductase activity, respectively[37,38]. However, most of the factors implicated in Cyt$b_6f$ accumulation

act in co-factor integration, and very little is known about the pathway governing the assembly of the protein subunits of the Cyt$b_6f$ complex.

In the present work, we have identified and characterized an essential assembly factor for Cyt$b_6f$ complex biogenesis in Arabidopsis. The protein, encoded by locus AT2G27290, facilitates the assembly of the core protein subunits of the complex. The corresponding gene was found in a time-resolved analysis of gene expression during the greening of etiolated tobacco (*Nicotiana tabacum*) plants[39]. The gene, referred to as de-etiolation-induced protein 1 (DEIP1), responded very early to the light signal by showing pronounced induction of mRNA accumulation. We show that knock-out mutants of *DEIP1* are incapable of photo-autotrophic growth and almost completely lack fully assembled Cyt$b_6f$ complexes. Blue-native polyacrylamide (PAA) gel electrophoresis (BN-PAGE), co-immunoprecipitation and bimolecular fluorescence complementation (BiFC) assays revealed that DEIP1 interacts with the Cyt$b_6f$ subunits PetA and PetB. The *deip1* mutants exhibit altered accumulation of Cyt$b_6f$ complex intermediates, suggesting a role in the initial steps of Cyt$b_6f$ complex assembly. The identification of DEIP1 provides an entry point into disentangling the assembly process of the Cyt$b_6f$ complex that previously has not been amenable to experimental investigation.

## Results

### DEIP1 is a thylakoid-anchored protein essential for photosynthesis and autotrophic growth.

To identify new regulators of light-induced chloroplast biogenesis, we screened our previous highly time-resolved RNAseq datasets of greening tobacco leaves[39] for genes whose transcripts responded early to the de-etiolation signal. One of the earliest light-induced genes of unknown function was locus 104247735, identified in the nuclear genome of the diploid progenitor species *Nicotiana sylvestris* of the allotetraploid tobacco, *Nicotiana tabacum*[40]. Expression of the locus showed pronounced induction already after 60 min of illumination and peaked as early as after 120 min (Supplementary Fig. 1a). We preliminarily named the corresponding putative gene product de-etiolation induced protein 1 (DEIP1). Its putative ortholog in Arabidopsis is encoded by locus AT2G27290 and shows high expression in all green tissues, including cotyledons and true leaves (Supplementary Fig. 1b)[41,42]. The predicted protein comprises 201 amino acids, and contains a predicted chloroplast transit peptide (according to the SUBA4 database[43]) at the N-terminus, an annotated domain of the uncharacterized function, DUF1279, at the C-terminus, and a single predicted transmembrane segment (Supplementary Fig. 1c)[44]. The DUF1279 of DEIP1 is conserved in all embryophytes, and a putative ortholog is also found in the green alga *Chlamydomonas reinhardtii* (Supplementary Figs. 2, 3; Supplementary Table 1). However, no homologs of DEIP1 were found in cyanobacteria.

De-etiolation experiments conducted with Arabidopsis seedlings germinated in the dark revealed that, similar to tobacco, *DEIP1* transcripts are strongly upregulated during light-induced greening, displaying a 12-fold induction 48 h after the onset of illumination (Supplementary Fig. 4a). The light-induced expression of *DEIP1* followed a similar kinetics as the *LHCB2* transcript (encoding a component of the PSII light-harvesting antenna), while mRNA accumulation of *LPOR* (encoding the light-dependent protochlorophyllide oxidoreductase) showed a rapid decrease within the first 6 h of light (Supplementary Fig. 4a), as previously reported[45].

To address the physiological function of the DEIP1 gene product, collections of T-DNA insertion lines were screened for mutants potentially affecting locus AT2G27290. Two potential

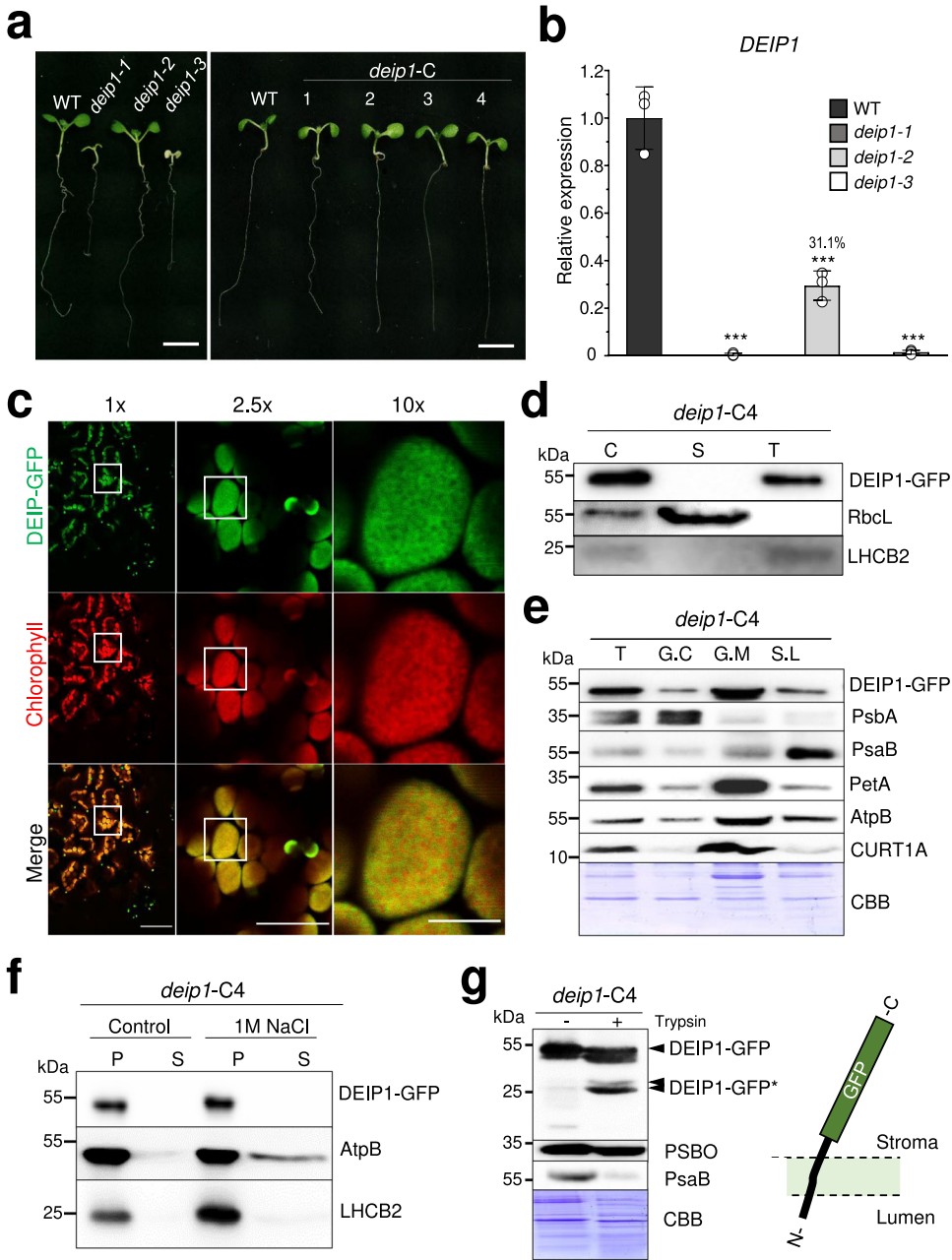

mutant alleles were identified and the predicted insertions were experimentally verified. One of the mutants harbors the T-DNA insertion in the first intron (*deip1-1*; SALK_ 037263), whereas the second mutant carries it in the 5′ untranslated regions (5′ UTR; *deip1-2*; SALK_067267C; Supplementary Fig. 4b, c). Since T-DNA insertions in introns and UTRs often produce leaky mutants that do not represent true null alleles, we used CRISPR/Cas9-based genome editing to additionally produce genuine knock-out mutants. A genome-edited line harboring a large deletion of 1084 bp in the coding region of AT2G27290 was selected for further analysis (*deip1-3*; Supplementary Fig. 4b–d).

Homozygous *deip1-1* and *deip1-3* mutants displayed striking pigment-deficient phenotypes and were incapable of photoautotrophic growth under moderate light (95 μmol m$^{-2}$ s$^{-1}$) conditions, thus necessitating their maintenance as heterozygous plants (Fig. 1a, Supplementary Fig. 5a). By contrast, the T-DNA insertion line *deip1-2* resembled the wild type (Fig. 1a). To correlate the observed phenotypes with *DEIP1* gene expression,

the transcript accumulation was measured by qRT-PCR analysis. The data revealed that, while *DEIP1* mRNA accumulation was reduced to undetectably low levels in mutant lines *deip1-1* and *deip1-3*, the *deip1-2* mutant accumulated the mRNA to ~30% of the wild-type level (Fig. 1b). These results suggest that both the T-DNA insertion into intron 1 (*deip1-1*) and the CRISPR/Cas9 deletion of the coding region (*deip1-3*) represent true knock-out alleles, while the *deip1-2* mutant represents a mild knock-down allele (Supplementary Fig. 4b; Fig. 1b) that may not appreciably affect DEIP1 function. This interpretation gained further support from chlorophyll content measurements that revealed strongly reduced chlorophyll contents in the *deip1-1* and *deip1-3* mutants, but not in the *deip1-2* mutant (Supplementary Fig. 5b).

To ultimately confirm that DEIP1 loss of function was causally responsible for the observed mutant phenotypes and, at the same time, be able to determine the subcellular and suborganellar localization of the protein, a transformation constructs for genetic complementation was generated by fusing the coding sequence of

**Fig. 1 Isolation of *deip1* mutants, genetic complementation, subcellular localization analysis of the DEIP1 protein, and thylakoid protein accumulation in wild-type and mutant plants. a** Mutant phenotypes. 7-day-old seedlings of the wild type (WT), the *deip1-1*, *deip1-2*, and *deip1-3* mutants, and the four complemented lines *deip1*-C1-4 were raised under a moderate light intensity of 95 µmol m$^{-2}$ s$^{-1}$. Scale bars: 5 mm. $n = 6$ independent biological replicates. **b** *DEIP1* transcript accumulation in the Arabidopsis *deip1-1*, *deip1-2*, and *deip1-3* mutants relative to wild-type plants (WT). The qRT-PCR data for the mutants were normalized and compared to the wild type. $n = 3$ biological replicates; error bars indicate standard deviation; Statistical significance was determined by one-way ANOVA, P values were adjusted by Tukey post-test for multiple comparisons, ***$P < 0.001$. **c** Localization of the DEIP1-GFP fusion protein as analyzed by confocal laser-scanning microscopy. The green fluorescence of DEIP1-GFP (green; upper panels), the chlorophyll fluorescence (red; middle), and the merged images (bottom) are shown. Scale bars: 50 µm (left panels), 20 µm (middle panels) and 5 µm (right panels). $n = 3$ independent biological replicates. **d** Immunoblot analysis of DEIP1-GFP localization in complemented line *deip1*-C4. Samples of 10 µg total chloroplast protein (C), and purified stroma (S) and thylakoid fractions were separated in a 12.5% PAA gel, and probed with anti-GFP, anti-RbcL, and anti-LCHB2 antibodies. $n = 3$ independent biological replicates. **e** Localization of DEIP1-GFP in intact thylakoids (T), and in fractionated thylakoids separated into grana core (G.C), grana margins (G.M), and stroma lamellae (S.L). Protein amounts equivalent to 1 µg chlorophyll were loaded for each fraction, and the blots were hybridized to an anti-GFP antibody for detection of the DEIP1-GFP fusion protein. The quality of the fractionation was assessed with antibodies against PsbA (highly abundant in G.C), PsaB (highly abundant in S.L), PetA (primarily present in G.M, but also in G.C and S.L), AtpB (primarily present in GM and SL), and CURT1A (highly abundant in GM). The Coomassie-stained PAA gel (CBB) is shown below the immunoblot. $n = 3$ independent biological replicates. **f** Salt wash of thylakoid samples from *deip1*-C4. Thylakoid samples equal to 1 µg of chlorophyll were incubated with 1 M NaCl for 1 h and separated in pellet (P) and soluble (S) fractions by centrifugation. Proteins present in the pellet and soluble fractions were resolved in a 12% PAA gel, and probed with anti-GFP, anti-AtpB, and anti-LHCB2 antibodies. $n = 3$ independent biological replicates. **g** Trypsin protection assay. Thylakoids were isolated from *deip1*-C4, protein samples equivalent to 2 µg of chlorophyll were incubated with 1 µg mL$^{-1}$ trypsin at 37 °C for 30 min, resolved in a 12.5% PAA gel, and probed with anti-GFP, anti-PSBO (detecting a marker protein that resides at the lumenal side of the membrane and, therefore, is protected from tryptic digestion), and anti-PsaB antibodies (recognizing a protein containing domains exposed to the stromal side and, therefore, being susceptible to tryptic digestion). The DEIP1-GFP degradation products resulting from tryptic digestion are indicated as DEIP1-GFP*. As a control for equal loading, the Coomassie-stained PAA gel (CBB) is shown below the immunoblot. The cartoon illustrates the topology of the DEIP1-GFP protein in the thylakoid membrane. $n = 2$ independent biological replicates.

*DEIP1* to a C-terminal GFP tag. The construct was introduced into heterozygous *deip1-1* mutant lines by *Agrobacterium*-mediated transformation, followed by isolation of transgenic lines that were homozygous for both the *deip1-1* T-DNA insertion and the P$_{UBQ10}$::DEIP1-GFP construct by genetic segregation (Supplementary Fig. 6a). Four independent doubly homozygous complemented lines were isolated (subsequently referred to as *deip1*-C1 to *deip1*-C4). Transcript accumulation from the *DEIP1-GFP* fusion gene in the complemented lines was 20 to 80 times higher than that from the native *DEIP1* locus in wild-type plants (Supplementary Fig. 6b), and accumulation of the DEIP1-GFP fusion protein was readily detectable by immunoblot analysis (Supplementary Fig. 6c). Full complementation of the *deip1-1* mutant phenotype was observed in all four lines (Fig. 1a), including restoration of the chlorophyll content to wild-type levels (Supplementary Fig. 6d). Analysis of GFP fluorescence by confocal laser-scanning microscopy revealed chloroplasts localization of DEIP1-GFP (Fig. 1c), consistent with the predicted presence of an N-terminal transit peptide for import into plastids. Immunoblot analysis of soluble and thylakoid fractions showed that DEIP1 is associated with thylakoids (Fig. 1d). To resolve the localization within thylakoids in greater detail, thylakoid preparations were fractionated into grana cores, grana margins and stroma lamellae (Fig. 1e). Immunoblot analyses using an anti-GFP antibody (for detection of the DEIP1-GFP fusion) and a set of marker proteins as controls revealed that DEIP1 predominantly localizes to the grana margins. Its distribution in the various thylakoid fractions was strikingly similar to that of PetA, the cytochrome *f* subunit of the Cyt*b$_6$f* complex (Fig. 1e). Washes of isolated thylakoid membranes with 1 M NaCl showed that DEIP1 is strongly anchored to the membrane (Fig. 1f) and suggested that it is a membrane-spanning protein. To test this assumption and determine the topology of the protein in the thylakoid membranes, trypsin digestion assays were performed. Protease susceptibility of the GFP tail indicated that the C-terminus of DEIP1 is exposed to the stromal side, while the N-terminus is likely exposed to the lumenal side of the thylakoid membrane (Fig. 1g).

Next, we explored the role of DEIP1 in photosynthesis by pulse amplitude modulation (PAM) measurements in the wild-type and the *deip1-1*, *deip1-2* and *deip1-3* mutants (Fig. 2). The maximum quantum yield of photosystem II (F$_v$/F$_m$) and the effective quantum yield (Φ$_{II}$) were found to be strongly reduced under both low-light (LL; 56 PAR) and high-light (HL; 461 PAR) conditions in the *deip1-1* and *deip1-3* mutants (Fig. 2a). While none of the mutants showed pronounced defects in non-photochemical quenching (NPQ) under low light, NPQ was much reduced in the *deip1-1* and *deip1-3* mutants under high light (Fig. 2b). Finally, the chlorophyll fluorescence parameter qL (reflecting the fraction of PSII reaction centers that are 'open', with the primary quinone-type acceptor Q$_A$ being in the oxidized state) and the electron transport rate (ETR) were severely reduced in *deip1-1* and *deip1-3* under both low light and high light, and mildly reduced in *deip1-2* under high light (Fig. 2c, d). Next, we measured chlorophyll *a* fluorescence induction kinetics in the wild-type and the *deip1* mutants (Fig. 2e). While the fluorescence curves were similar in the wild-type and the *deip1-2* mutant, the curves for *deip1-1* and *deip1-3* showed the typical pattern occurring when electron transfer from PSII to downstream components of the photosynthetic electron transport chain is impaired (Fig. 2e)[46]. As expected, all measured photosynthetic parameters were restored to wild-type-like levels in the DEIP1-GFP complemented lines (Supplementary Fig. 7).

The *deip1-2* mutant displayed no visible phenotype upon growth under standard conditions (Fig. 1a), but showed mild defects in qL and ETR under high light (Fig. 2c, d). This observation prompted us to examine the photosynthetic performance of this mutant during acclimation from moderate light (95 µmol m$^{-2}$ s$^{-1}$) to high-light conditions (600 µmol m$^{-2}$ s$^{-1}$; Supplementary Fig. 8). We observed a pronounced decrease in F$_v$/F$_m$ in cotyledons of *deip1-2* upon exposure to high light (Supplementary Fig. 8a), but not under standard light conditions. However, chlorophyll contents did not differ significantly between the wild type and the *deip1-2* mutant in any of the tested conditions (Supplementary Fig. 8b).

Finally, we examined morphology and ultrastructure of *deip1-1*, *deip1-2*, and *deip1-3* mutant chloroplasts (Fig. 3). Confocal laser-scanning microscopy revealed that mesophyll cells of the two knock-out mutants, *deip1-1*, and *deip1-3*, harbored smaller chloroplasts than cells of the wild-type and the mild knock-down

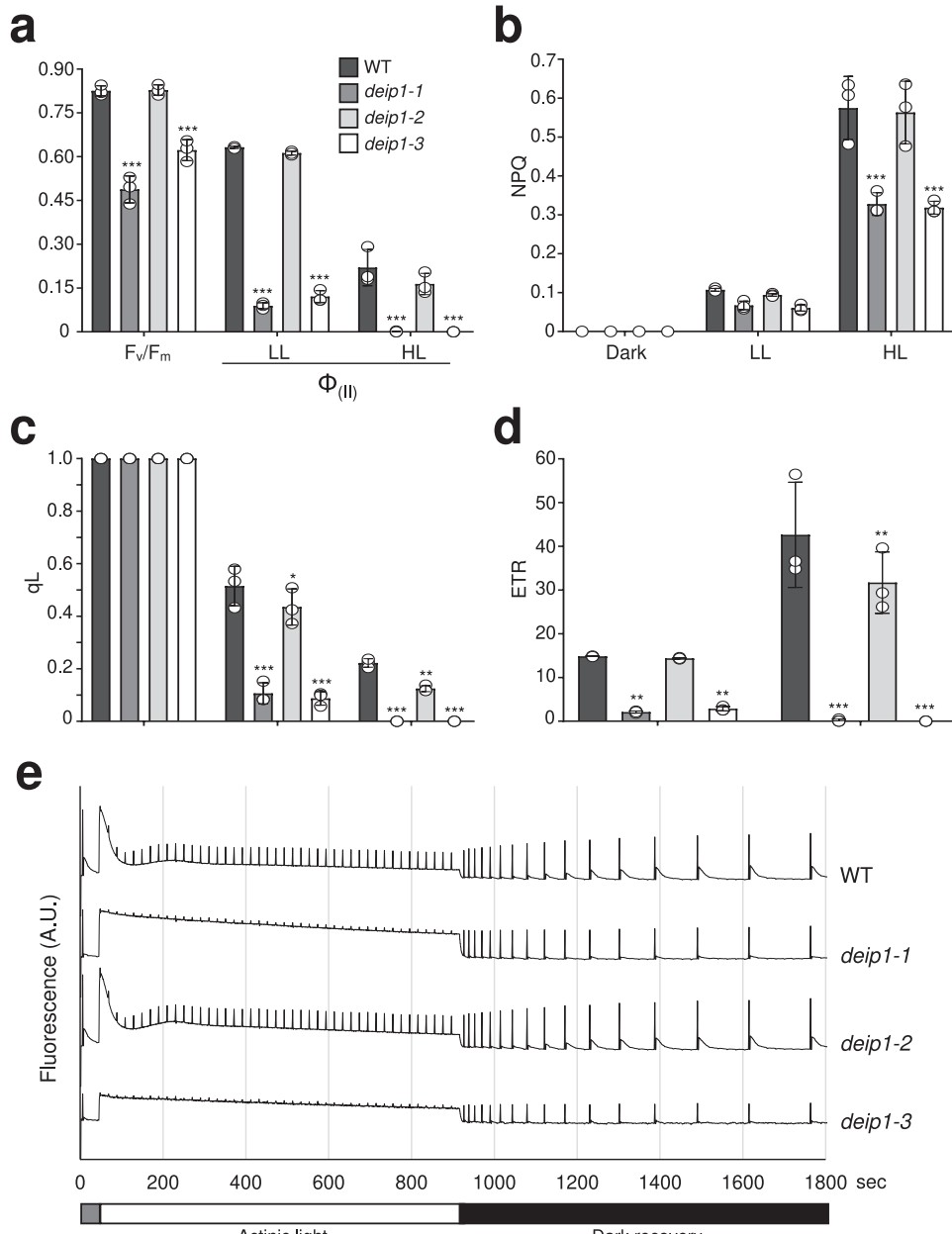

**Fig. 2 Measurement of photosynthetic performance in the wild-type and the *deip1-1, deip1-2,* and *deip1-3* mutants.** The chlorophyll fluorescence parameters **a** maximum quantum yield ($F_v/F_m$) and effective quantum yield ($\Phi_{II}$), **b** non-photochemical quenching (NQP), **c** open reaction centers of photosystem II (qL), and **d** electron transport rate (ETR) were obtained by PAM imaging of wild-type (WT), *deip1-1, deip1-2,* and *deip1-3* seedlings grown for 7 days. $n = 3$ independent biological replicates (with 15 seedlings each); error bars represent standard deviation; Statistical significance was determined by one-sided two-way ANOVA, *P* values were adjusted by Tukey post-test for multiple comparisons and comparing the *deip1* mutants to the wild type at the same light intensity; *$P < 0.05$, **$P < 0.01$, ***$P < 0.001$. LL: low light (56 PAR); HL: high light (461 PAR). **e** Chlorophyll *a* fluorescence induction kinetics of wild-type plants (WT), *deip1-1, deip1-2,* and *deip1-3*. The intensity of the actinic light was 198 PAR. The curves are representative measurements of 8–15 seedlings ($n = 3$).

mutant *deip1-2* (Fig. 3a). However, the typical dotted fluorescence pattern originating mainly from LHCII complexes residing in grana stacks[47] was still detectable in the knock-out mutants. Further analysis of thylakoid architecture by transmission electron microscopy (TEM; Fig. 3b) confirmed the smaller chloroplast size in the *deip1-1* and *deip1-3* mutants. Although grana stacking was observed in all three mutants, aberrant membrane splits were seen near the grana stacks, and also some discontinuous membranes in the non-appressed thylakoid regions (Fig. 3b).

Taken together, our characterization of the mutant phenotypes demonstrates that DEIP1 is essential for photoautotrophic growth, localizes to the grana margins, resides within the membrane, and is required for photosynthesis and proper chloroplast development.

**DEIP1 is required for the accumulation of the Cyt$b_6f$ complex, but does not contribute to synthesis or degradation of its subunits.** To test whether the pale phenotypes observed in the knock-out mutants *deip1-1* and *deip1-3* are the results of photo-

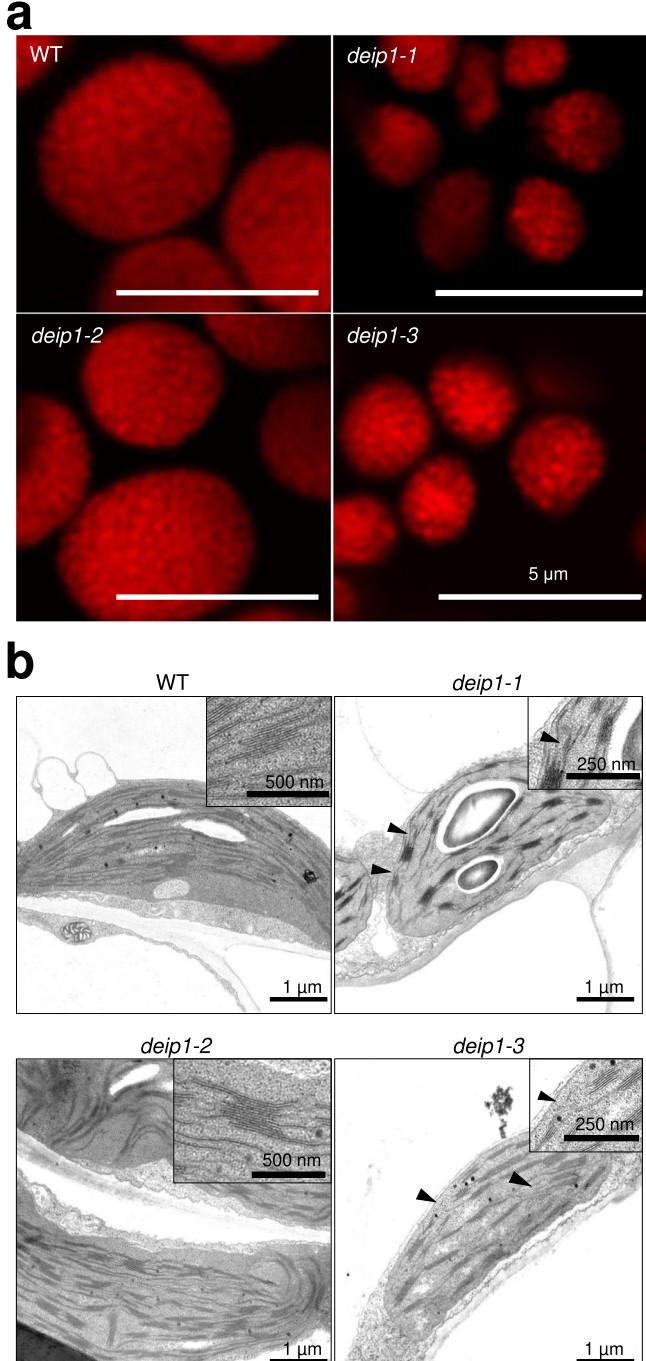

**Fig. 3 Confocal laser scanning microscopy (CLSM) and transmission electron microscopy (TEM) images of chloroplasts in the wild type (WT) and the *deip1-1*, *deip1-2*, and *deip1-3* mutants. a** CLSM images of chloroplasts. The red fluorescence corresponds to the chlorophyll fluorescence in chloroplasts of mesophyll cells, the dotted pattern reflects the arrangement of grana stacks. Representative images were selected based on observation of 50 chloroplasts per genotype. $n = 2$ independent biological replicates. **b** TEM micrographs obtained from chloroplasts of mesophyll cells. The arrowheads mark fork-like structures close to appressed membranes within thylakoids of *deip1-1* and *deip1-3*. Representative images were selected based on observation of 15 chloroplasts. $n = 2$ independent biological replicates.

oxidative damage, plants were grown for 5 weeks under extreme low-light conditions (5 µmol m$^{-2}$ s$^{-1}$) in heterotrophic culture. Indeed, the mutants were visibly indistinguishable from wild-type plants (Fig. 4a) under these conditions, and chlorophyll contents

reached wild-type levels (Supplementary Fig. 9a). This finding raised the possibility that the light sensitivity of the *deip1* mutants is due to an imbalance in the photosynthetic electron transfer chain, which causes photo-oxidative stress and triggers rapid destruction of the protein complexes in the thylakoid membrane. We, therefore, investigated the accumulation of the major thylakoidal protein complexes in total protein extracts from wild-type and mutant seedlings by immunoblot analyses (Fig. 4b). When seedlings were grown in moderate light, we observed complete absence of the Cyt$b_6f$ subunits PetA and PetB, and reduced accumulation of subunits of PSII, PSI, and cpATP synthase (to ~25%) in the *deip1-1* and *deip1-3* knock-out mutants, while the *deip1-2* mutant displayed a reduction in Cyt$b_6f$ subunit accumulation to ~50–75% of the wild-type level and no reduction in the other protein complexes (Fig. 4b). Based on these patterns, we suspected that the primary defect in the *deip1* mutants is in Cyt$b_6f$, and the disruption of the photosynthetic electron transfer chain by the Cyt$b_6f$ deficiency causes severe photo-oxidative damage that secondarily results in lower accumulation of all other thylakoidal protein complexes. To test this assumption, protein accumulation was investigated in plants grown for 5 weeks under extreme low-light conditions (5 µmol m$^{-2}$ s$^{-1}$), where no phenotypic differences were observed between the wild type and the mutants. These immunoblot analyses revealed that the Cyt$b_6f$ subunits are virtually undetectable in the *deip1-1* and *deip1-3* mutants (Fig. 4b; Supplementary Fig. 9b), whereas the subunits of the other protein complexes accumulated to wild-type-like levels (Fig. 4b). To verify that the accumulation patterns observed in total protein samples faithfully reflect the composition of the thylakoid membranes, we also examined the abundance of Cyt$b_6f$ subunits in thylakoids purified from 7-day-old seedlings grown under moderate light and 5-week-old plants grown under low light by immunoblot analyses. These experiments confirmed the absence of all major Cyt$b_6f$ subunits (PetA, PetB, PETC, and PetD) from the *deip1* knock-out mutants in moderate light, and their extremely low accumulation in low light (Fig. 4c). When the complemented lines were analyzed, we observed wild-type levels of PetA and PETC in all *deip1*-C lines, suggesting that the overaccumulation of DEIP1 in these plants does not result in excessive accumulation of the Cyt$b_6f$ complex (Supplementary Fig. 9c).

Next, we analyzed the main photosynthetic complexes by blue-native polyacrylamide (PAA) gel electrophoresis (BN-PAGE) of thylakoid samples from wild-type seedlings and the *deip1-1*, *deip1-2*, and *deip1-3* mutants grown under moderate light (Fig. 5a, b). These experiments confirmed the absence of the Cyt$b_6f$ complex in the knock-out mutants and revealed decreased abundance of PSII supercomplexes, the PSI/PSII dimer band and the LHCII assembly in *deip1-1* and *deip1-3* grown under moderate light (Fig. 4a). Subsequent 2D-SDS-PAGE further confirmed the absence of the Cyt$b_6f$ complex from both *deip1-1* and *deip1-3*, and decreased complex abundance in the *deip1-2* mutant (Supplementary Fig. 10). In agreement with our immunoblot analyses (Fig. 4c), the PSII and PSI complexes accumulated to wild-type-like levels in *deip1* knock-out mutants grown under low light (Fig. 5b), while no detectable accumulation of the Cyt$b_6f$ complex occurred.

The function of DEIP1 in the chloroplast and its requirement for Cyt$b_6f$ accumulation would be compatible with a role in the expression of plastid-encoded Cyt$b_6f$ subunits, or alternatively, a function in complex assembly and/or stability. To test for a possible function in transcription, mRNA turnover or translation, we conducted array-based RNA and ribosome profiling in wild-type plants and the *deip1-1* knock-out mutant grown under extremely low light, where the effects of photo-oxidative damage are minimal (Fig. 5c, d; Supplementary Fig. 11; Supplementary Data 1). The biological replicates showed high reproducibility in

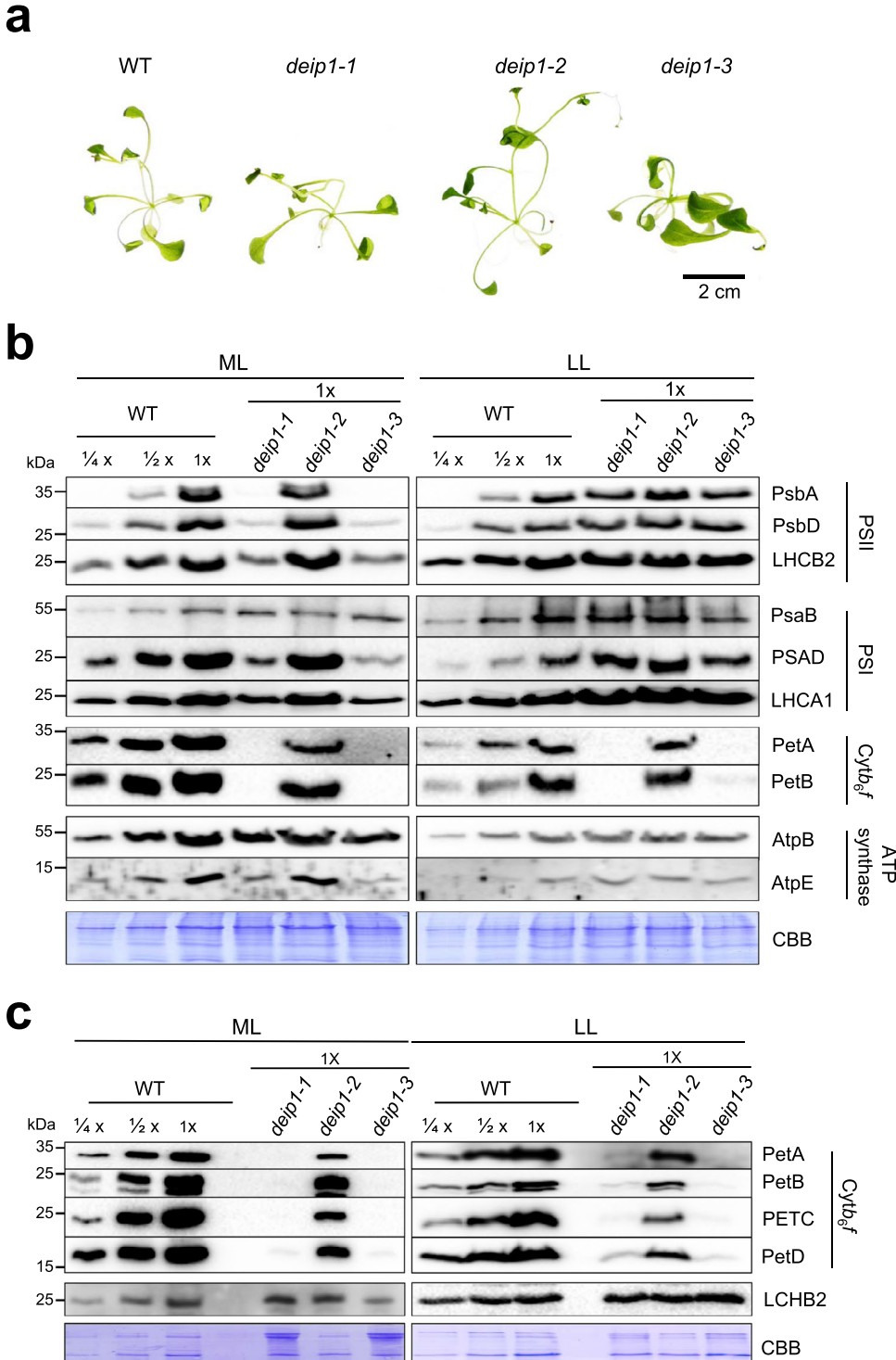

**Fig. 4 Light-dependent accumulation of thylakoid proteins in *deip1* mutants. a** Phenotypes of 5-week-old plants grown in vitro under extreme low-light conditions (5 μmol m$^{-2}$ s$^{-1}$). **b** Accumulation of diagnostic protein subunits of the major thylakoidal protein complexes in the wild type (WT), and the *deip1-1*, *deip1-2*, and *deip1-3* mutants grown under moderate light (ML; 95 μmol m$^{-2}$ s$^{-1}$) or extreme low light (LL; 5 μmol m$^{-2}$ s$^{-1}$). Samples of 20 μg total protein were loaded for *deip1-1*, *deip1-2*, and *deip1-3* (1×), and blots were probed with antibodies against proteins of the PSII complex (PsbA, PsbD, and LHCB2), the PSI complex (PsaB, PSAD and LCHA2), the Cyt$b_6f$ complex (PetA, PetB) and the chloroplast ATP synthase (AtpB and AtpE). As a control for equal loading, the Coomassie-stained PAA gel (CBB) is shown below each series of immunoblots. $n = 3$ independent biological replicates. **c** Protein accumulation in thylakoids purified from the wild type (WT) and the *deip1* mutants grown under moderate light (ML) or extreme low light (LL). Thylakoid protein samples equivalent to 1 μg of chlorophyll were resolved in 12.5% PAA gels and blotted. Immune detection was conducted with antibodies against the Cyt$b_6f$ subunits PetA, PetB, PETC, and PetD. The light-harvesting complex protein LHCB2 served as a marker for equal chlorophyll-based loading, and the Coomassie-stained PAA gel (CBB) is shown below each series of immunoblots. Note that, due to their low chlorophyll content, the amount of loaded protein is substantially higher in the *deip1-1* and *deip1-3* mutants grown under ML. $n = 3$ independent biological replicates.

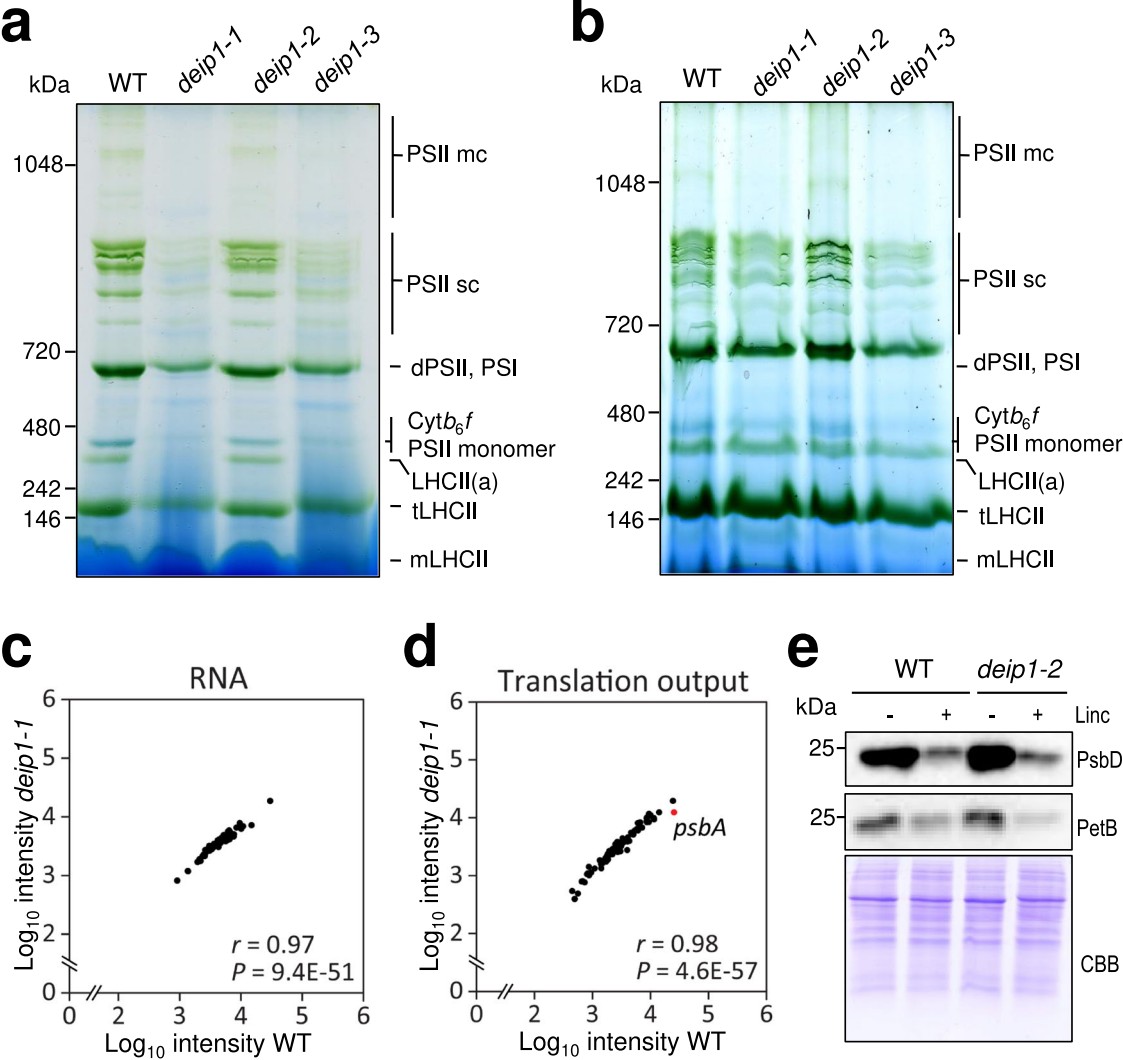

**Fig. 5 DEIP1 is involved in the accumulation of Cyt$b_6f$, but does not contribute to subunit expression or complex stability. a** BN-PAGE of thylakoids from the wild type (WT) and the *deip1-1*, *deip1-2*, and *deip1-3* mutants grown under moderate light intensity. Thylakoid samples equivalent to 8 µg of chlorophyll were solubilized with the non-ionic detergent β-DDM, and resolved in a 6-12.5% native gradient gel. The main photosynthetic complexes are indicated as PSII megacomplexes (PSII mc), PSII supercomplexes (PSII sc), Cyt$b_6f$, LHCII assembly (LHCII(a)), LHCII trimer (tLHCII) and monomeric LHCII (mLHCII). The molecular weights (in kDa) of the marker protein bands are given at the left of the gels. $n = 3$ independent biological replicates. **b** BN-PAGE of thylakoid samples from the wild type (WT) and the *deip1-1*, *deip1-2* and *deip1-3* mutants grown under low light. Thylakoid samples equal to 8 µg of chlorophyll were solubilized with β-DDM, and resolved in a 6–12.5% native gradient gel. The main photosynthetic complexes are indicated as PSII megacomplexes (PSII mc), PSII supercomplexes (PSII sc), PSII dimer and PSI, Cyt$b_6f$, LHCII assembly (LHCII(a)), LHCII trimer (tLHCII) and monomeric LHCII (mLHCII). The molecular weight (in kDa) of the marker protein bands are given at the left of the gels. $n = 3$ independent biological replicates. **c** Transcript levels (RNA) and **d** translation output (ribosome footprint abundances) as analyzed by microarray-based ribosome profiling of *deip1-1* and wild-type seedlings grown under low light. Average ribosome footprint and transcript abundances in *deip1-1* were calculated for each chloroplast reading frame from three biological replicates, and the mean signal intensities were plotted against the wild type. The labeled *psbA* reading frame represents the only chloroplast-encoded gene with a significant ($P < 0.05$) and more than 2-fold change in expression. Statistical significance was determined by LIMMA, $P$ values were adjusted by FDR-test for multiple comparisons and compared to the wild type (Supplementary Data 1). **e** Immunoblot of total protein extracts from the wild type (WT) and the *deip1-2* mutant 4 days after infiltration with water (−) or 1 mM lincomycin (Linc; +). The turnover of the Cyt$b_6f$ complex was assessed by immunoblotting with an anti-PetB antibody. The anti-PsbD antibody was used as a positive control for the effectiveness of the lincomycin treatment, and the Coomassie-stained PAA gel (CBB) is shown as loading control. $n = 3$ independent biological replicates.

ribosome footprints and RNA accumulation data across the plastid genome (with average *r*-values > 0.99 for both transcriptomic and ribosome profiling data; Supplementary Data 1). No pronounced changes in transcript accumulation were observed between the wild-type and the *deip1-1* mutant (Fig. 5c). However, we observed a significantly (>2-fold) lower ribosome footprint abundance for the *psbA* mRNA (2.3-fold) in *deip1-1* compared to the wild type (Fig. 5d; Supplementary

Fig. 11), which appears to be caused by a reduced translation efficiency of unaltered *psbA* transcript levels. Translation of *psbA* is known to be the step in plastid gene expression that is most sensitive to environmental changes and genetic perturbations (e.g., changes in light intensity[48]), suggesting that its down-regulation in the *deip1* knock-out plants represents a response to the Cyt$b_6f$ deficiency (Fig. 5d). Importantly, no differences in the translation output or transcript accumulation for chloroplast-

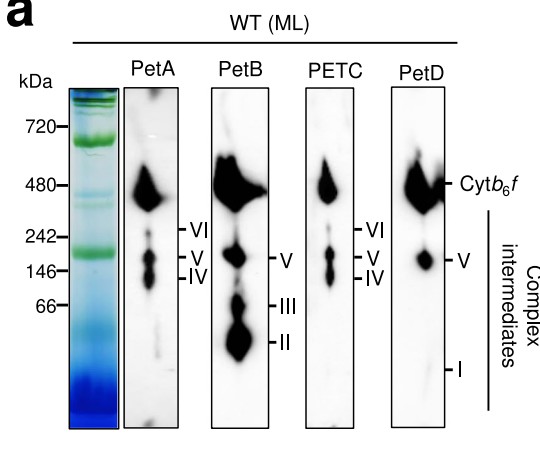

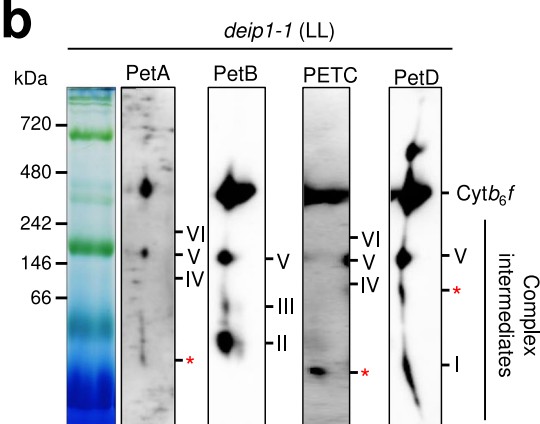

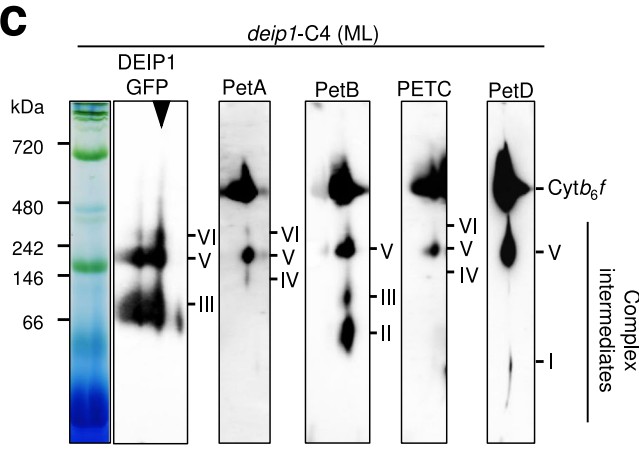

**Fig. 6 DEIP1 co-migrates with core subunits of the Cyt$b_6f$ complex.** 2D-SDS-PAGE analyses of thylakoids purified from **a** wild-type plants grown under moderate (ML) light for 7 days, **b** *deip1-1* mutants grown under low light (LL) for 5 weeks, and **c** the *deip1*-C4 complemented line grown under moderate light (ML) for 7 days. Thylakoid samples equivalent to 20 µg chlorophyll were solubilized in β-DDM, and resolved in an 8–13.8% gradient gel. SDS-PAGE was performed as second dimension, and immunodetection was conducted with anti-PetA, anti-PetB, anti-PETC, and anti-PetD antibodies. For the *deip1*-C4 line, the antibody against GFP was included, and the band that corresponds to DEIP1-GFP is indicated by the black arrowhead. Putative assembly intermediates are indicated as I, II, III, IV, V, and VI. See text for details. *n* = 3 independent biological replicates.

encoded Cyt$b_6f$ genes (or any other chloroplast gene besides *psbA*) were seen in the mutant, indicating that DEIP1 does not act at the level of Cyt$b_6f$ subunit expression in the chloroplast.

The potential involvement of DEIP1 in Cyt$b_6f$ complex stability was assessed by blocking de novo synthesis with the chloroplast translation inhibitor lincomycin and following complex accumulation over time (Fig. 5e). Due to the long lifetime of the Cyt$b_6f$ complex[49], and to avoid extreme photo-damage due to the lack of D1 turnover[50], the seedlings were infiltrated with lincomycin (or water as control) and incubated in the dark at 4 °C for 16 h, and then transferred to liquid media for 4 days under low light intensities. Immunoblot analysis showed that the decline in the contents of the Cyt$b_6f$ subunit PetB and the PSII subunit PsbD (analyzed as a control) was similar in lincomycin-infiltrated wild-type leaves and *deip1-2* mutant leaves, suggesting that DEIP1 does not act in the degradation or stabilization of Cyt$b_6f$ subunits (Fig. 5e).

**DEIP1 interacts with PetA and PetB and acts as an assembly factor for the Cyt$b_6f$ complex.** To further explore the role of DEIP1 in the accumulation of Cyt$b_6f$ complex, we employed BN-PAGE and 2D-SDS-PAGE to analyze thylakoid samples purified from 7-day-old wild-type seedlings grown under moderate light and 5-week-old *deip1-1* plants grown under low light(Fig. 6a, b). In the wild type, several putative assembly intermediates of Cyt$b_6f$ could be detected. Based on co-migration of Cyt$b_6f$ subunits, these intermediate complexes include: (i) three complexes containing PetA and PETC (intermediates IV, V, and VI), (ii) three complexes containing PetB (intermediates II, III, and V), and (iii) two complexes containing PetD (intermediates I and V; Fig. 6a). Interestingly, the *deip1-1* mutant accumulated monomeric forms of both PetA and PetC, and intermediates IV and VI were not detected (Fig. 6b). In addition, the *deip1-1* mutant exhibited lower accumulation of the complex intermediate III (containing PetB), and accumulated an additional putative assembly intermediate containing PetD, which did not co-migrate with PetA, PetB, or PETC (Fig. 6b). Next, we assessed the assembly intermediates in thylakoid samples extracted from *deip1*-C4 seedlings (Fig. 6c). DEIP1-GFP co-migrated with PetB (in intermediate III), with the intermediate complex containing all four core subunits (intermediate V) and with the complex containing PetA and PETC (intermediate VI; Fig. 6c). DEIP1 does not co-migrate with the fully assembled Cyt$b_6f$ complex, providing further evidence against a role in complex stabilization.

To verify the association of DEIP1 with other thylakoid proteins in protein complexes, bis(sulfosuccinimidyl)suberate (BS[3])-mediated crosslinking of thylakoid fractions from the *deip1*-C4 complemented line and, as a control, stromal fractions of a transgenic line expressing a soluble chloroplast-targeted GFP (ctpGFP line) was performed (Fig. 7a). DEIP1 was found to be present in the thylakoids within high molecular weight complexes (hmwDEIP1-GFP; Fig. 7a).

To provide direct evidence for the protein-protein interactions suggested by the co-migration patterns of Cyt$b_6f$ subunits and DEIP1 (Fig. 7b), co-immunoprecipitation (co-IP) experiments were conducted (Fig. 7b; Supplementary Fig. 12a, b). The crosslinker dithiobis(succinimidyl propionate) (DSP) was applied to stabilize protein interaction in thylakoid membranes isolated from *deip1*-C4 and the wild type prior the solubilisation and co-IP. These experiments revealed strong co-IP enrichment of DEIP1-GFP with PetA and PetB and much weaker with PetD. No interaction was detected with PETC and all control proteins from

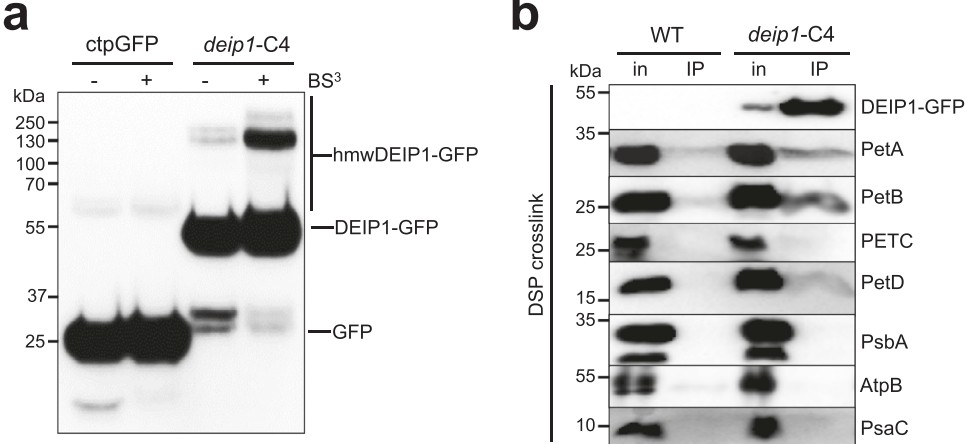

**Fig. 7 DEIP1 is associated in high molecular weight complexes and interacts primarily with PetA and PetB. a** BS[3]-mediated crosslink of stroma fractions and thylakoid samples obtained from transgenic plants expressing a soluble chloroplast-targeted GFP (ctpGFP) and the *deip1*-C4 line. The treated samples were separated in a 10% SDS PAA gel. An anti-GFP antibody was used to detect the soluble GFP, the monomeric DEIP1 (mDEIP1-GFP) and the high molecular weight complexes containing DEIP1 (hmwDEIP1-GFP). *n* = 3 independent biological replicates. **b** Co-IP from *deip1*-C4 and the wild type (WT). Thylakoid samples were treated with 1.5 mM DSP crosslinker prior to co-IP. The eluate was resolved by SDS-PAGE, and the immunodetection was done with anti-GFP, anti-PetA, anti-PetB, anti-PETC, anti-PetD, anti-PsbA, anti-AtpB and anti-PsaC antibodies. *n* = 3 independent biological replicates.

other thylakoidal protein complexes: the PSII subunit PsbA (as a protein with a high turnover rate), the ATP synthase subunit AtpB (as a protein enriched in grana margins), and the PSI subunit PsaC (Fig. 7b). The presence of weak signals for PetA, PetB, PetD, and AtpB in the co-IPs with wild-type samples is likely attributable to non-specific immunoprecipitation due to the crowding of proteins in the (highly protein-rich) thylakoid membranes[51].

To confirm the interactions of DEIP1 with Cyt$b_6f$ subunits by an independent method, bimolecular fluorescence complementation (BiFC) assays were performed[52] (Fig. 8). Interaction of DEIP1 with itself was observed (YFP fluorescence detected in 54% of the transfected protoplasts showing RFP fluorescence), suggesting that DEIP1 can undergo dimerization. Strong interactions were also seen with PetA (in 30% of the protoplasts) and with PetB (in 28% of the protoplasts; Fig. 8). No interaction was observed between DEIP1 and PETC, while the interaction with PetD was extremely weak and only observed in 17% of the protoplasts (Fig. 8). Overall, the results of the BiFC assays confirm the protein-protein interactions detected in our co-IP experiments and suggest that DEIP1 mediates essential steps in Cyt$b_6f$ assembly by interacting with the subunits PetA and PetB.

## Discussion

The Cyt$b_6f$ complex plays a central role in photosynthesis by functionally connecting PSII with PSI in the thylakoid membrane. In contrast to the two photosystems, very little is known about the biogenesis of Cyt$b_6f$ and the auxiliary factors involved in the Cyt$b_6f$ assembly process. In this work, we have identified and characterized a protein factor, DEIP1 that plays a crucial role in Cyt$b_6f$ assembly. Our data show that DEIP1 is a thylakoid-anchored protein that is specifically required for the biogenesis of Cyt$b_6f$. As photo-oxidative damage can secondarily lead to reduced levels of other thylakoidal protein complexes, we analyzed plant material grown under extreme low-light conditions. This approach minimizes photo-oxidative destruction of pigment-protein complexes in the thylakoid membrane of mutants with severe defects in the photosynthetic electron transport chain[14,27]. The results demonstrated that Cyt$b_6f$ is the only complex that is primarily affected in *deip1* loss-of-function mutants (Fig. 4). Consistent with this finding, the *deip1-2* knock-down mutant exhibits lower qL and ETR exclusively under high light

(Fig. 2c, d), a finding that can be attributed to the decreased activity of the Cyt$b_6f$ complex. As the Cyt$b_6f$ complex catalyzes the rate-limiting step in photosynthetic electron transfer, decreased complex activity in the mutants directly impacts lumenal acidification under high light and, in this way, photosynthetic control. Similar observations were made in the *pgr1* mutant (harboring a point mutation in PETC)[53], thus lending additional support to the conclusion that reduced Cyt$b_6f$ complex accumulation represents the primary defect in the *deip1* mutants.

The absence of Cyt$b_6f$ complex from *deip1-1* and *deip1-3* knock-out mutants resulted in smaller chloroplasts, and the thylakoid ultrastructure displayed aberrant fork-like splits, as commonly observed in immature thylakoids[54]. This observation confirms that Cyt$b_6f$ contributes to thylakoid maturation and shows that impaired thylakoid development negatively affects chloroplast development and size (Fig. 3). All *deip1* mutants formed grana stacks, reinforcing the notion that grana stacking is mainly, though not exclusively, mediated by the accumulation of LHCs[55].

The *DEIP1* gene was identified based on its very rapid induction upon reillumination of etiolated leaves (Supplementary Fig. 1). However, the Cyt$b_6f$ complex accumulates already in the prolamellar bodies (PLBs) of etioplasts[56], suggesting that the low-level basal expression of *DEIP1* is sufficient to facilitate the assembly of the Cyt$b_6f$ amounts required in etiolated seedlings. Clearly, the function of DEIP1 in the assembly of Cyt$b_6f$ in the dark warrants further study.

Our genome-wide analyses of mRNA accumulation, ribosome footprint abundance, and translation efficiency demonstrated that the expression of all chloroplast-encoded subunits of the Cyt$b_6f$ complex is unchanged in the *deip1* mutants, excluding a function of DEIP1 in the expression of complex subunits (Fig. 5c, d). However, we noted a decreased translation output of *psbA*, the gene encoding the D1 subunit of PSII. Similar changes in *psbA* expression were previously reported for other mutants with reduced Cyt$b_6f$ content[57], and therefore, likely represent secondary effects that are due to *psbA* being the most highly regulated chloroplast gene that is most sensitive to genetic perturbations and environmental changes[58].

The symmetric composition of the Cyt$b_6f$ dimer argues for formation of a monomeric Cyt$b_6f$ (containing all subunits) prior to formation of the functional Cyt$b_6f$ dimer. Complex assembly has

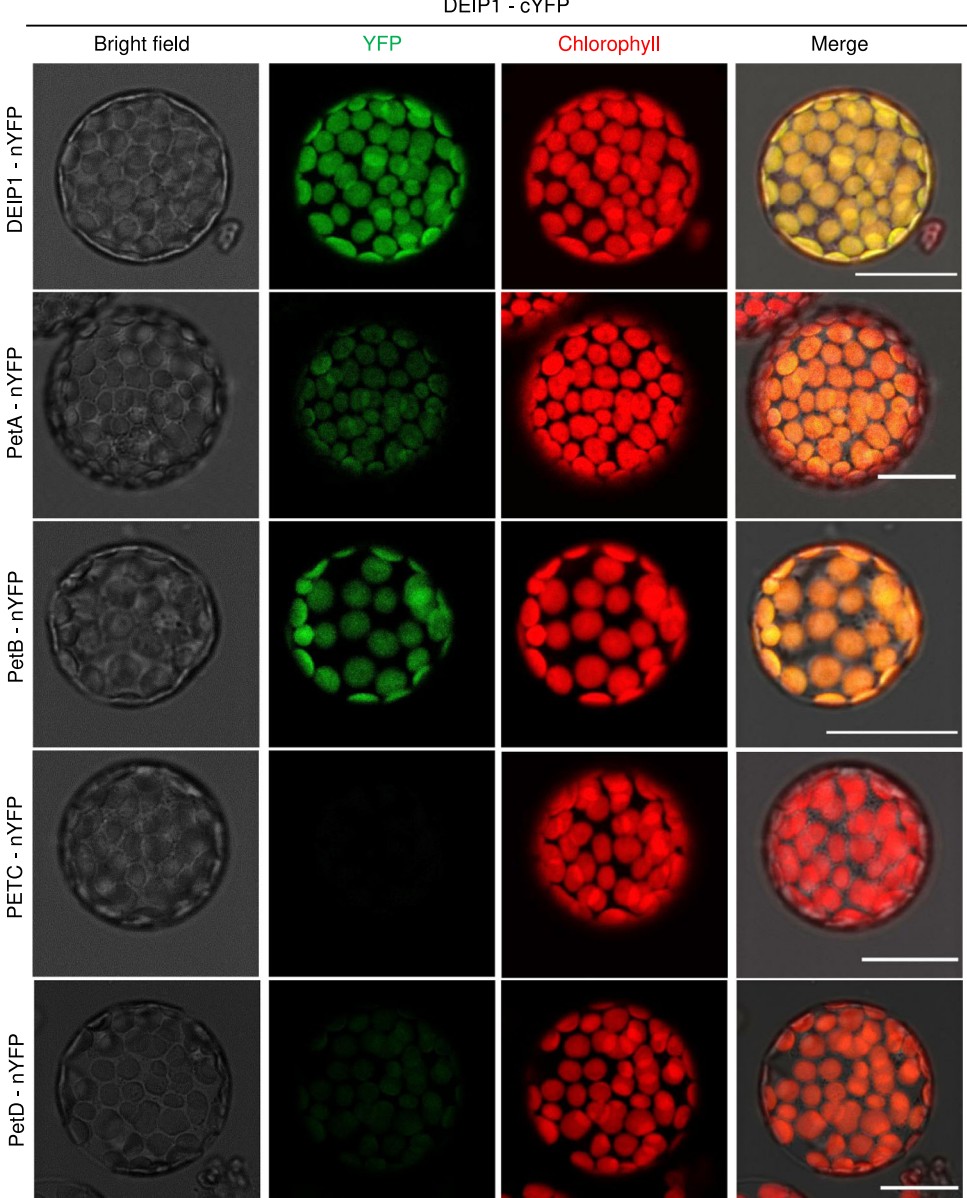

**Fig. 8 BiFC analyses to verify the interaction of DEIP1 with Cyt$b_6f$ complex subunits in chloroplasts.** Bimolecular fluorescence complementation (BiFC) assays with the DEIP1-cYFP fusion protein in tobacco protoplasts are shown. Interactions were assessed with the nYFP fused to the C-terminus of DEIP1, PetA, PetB, PETC and PetD. The endogenous transit peptide was employed for DEIP1, and the pea RBCS chloroplast transit peptide was employed for all Cyt$b_6f$ subunits. The bright field (BF), YFP (green), chlorophyll auto fluorescence (red) and the merged images are depicted. Representative images were selected based on the observation of 80 protoplasts in two independent experiments for each interaction. Scale bars: 20 μm.

been proposed to initiate with the membrane insertion of the four core subunits PetA, PetB, PETC and PetD (Fig. 9). Of the three chloroplast genome-encoded subunits, PetB and PetA are co-translationally inserted into the thylakoid membrane, while this may not be the case for PetD[59]. Previous in vitro experiments suggested that PetB and PetD are inserted into the thylakoid membrane by the signal recognition particle (SRP) pathway and the general insertase ALBINO3 (ALB3)[60,61]. In our two-dimensional gel electrophoresis experiments conducted to resolve putative assembly intermediates, we did not observe co-migration of PetB and PetD in low molecular weight complexes (assembly intermediate I containing PetD; assembly intermediates II and III containing PetB; Fig. 6a). Our data are consistent with a model, in which Cyt$b_6f$ assembly initiates with PetB and PetA insertion into the membrane prior to addition of PetD to PetB, and PETC to PetA (Fig. 9)[62,63]. In the wild type, none

of the four core subunits accumulate as a free protein, suggesting that the initial steps of complex assembly in the membrane occur very rapidly. This may include the association of PetB with auxiliary factors that remain to be identified (Fig. 6a; assembly intermediates II and III). Our BN-PAGE and two-dimensional gel electrophoresis suggested that DEIP1 accumulates as a dimer in the thylakoid membrane, and co-migrates with the PetB assembly intermediate III (Fig. 6c). In the absence of DEIP1, increased amounts of free and incompletely assembled PetD subunits accumulate (Fig. 6b), presumably because the PetB-DEIP1 interaction is required for formation of the PetB-PetD heterodimer (or other bottleneck effects that restrict assembly of the monomeric Cyt$b_6f$; Fig. 9).

Based on the intermediate complexes and the protein-protein interactions identified (Figs. 6–8), we propose that DEIP1 mediates the assembly of the monomeric Cyt$b_6f$ by interacting with

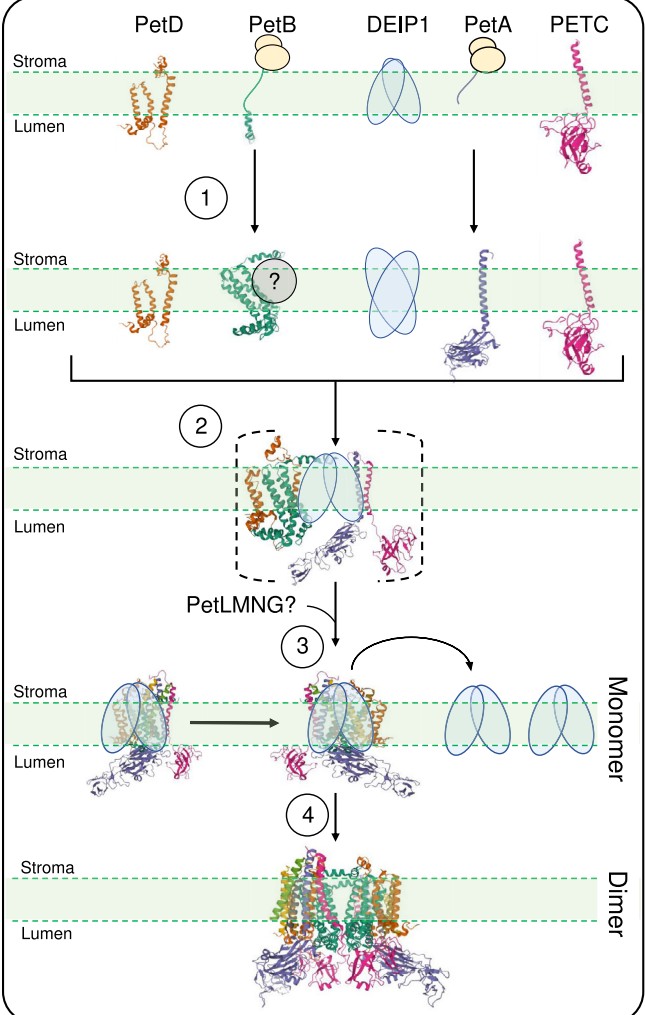

**Fig. 9 Model of the role of DEIP1 in the assembly of the Cytb₆f complex.** ① PetB and PetA are co-translationally inserted into the thylakoid membrane. Free DEIP1 is present as a homodimer. ② After its insertion into the membrane, PetD associates with PetB and DEIP1, and possibly other auxiliary factors. In parallel, PetA associates with PETC and DEIP1. The binding of DEIP1 to both heterodimers may bring the two subcomplexes in close proximity, thus facilitating their association. Transition of this unstable core complex to the stable monomeric Cytb₆f likely requires addition of the small subunits PetL, PetG, PetN, and PETM. ③ DEIP1 removal from the complex promotes the association of two monomers. ④ Formation of the stable dimeric Cytb₆f complex from two monomers completes the assembly process. See text for details. Images of the Cytb₆f complex and its subunits were obtained from rcsb.org[85] PDB ID 6RQF[25].

both the PetB-PetD and the PetA-PETC subcomplexes (assembly intermediate V; Figs. 6, 9). The abnormal accumulation of low molecular weight signals for PetA, PETC and PetD, and the decreased abundance of the PetB-containing assembly intermediate III in the *deip1-1* mutant (Fig. 6b) support the proposed critical role of DEIP1 in the assembly of the monomeric Cytb₆f complex (Fig. 9). Although it appears likely that the stabilizing small subunits PetG, PetL, PETM, and PetN are added only after the four core subunits have been assembled (Fig. 9)[26,28], the exact temporal order of their incorporation into the complex remains to be defined. Finally, it needs to be borne in mind that the Cytb₆f complex is tightly associated to lipids that influence complex assembly and stability[62,64]. It, therefore, cannot be excluded that some of the intermediate complexes and unassembled subunits

detected in the knock-out mutant (Fig. 6b) are the result of detergent treatments together with the complex-destabilizing effects of the absence of DEIP1.

Although the assembly of the Cytb₆f is largely unclear, our co-immunoprecipitation analyses and BiFC assays indicate that DEIP1 interacts primarily with PetA and PetB, but not with the PETC, and only weakly with PetD (Figs. 7, 8). These data suggest that DEIP1 acts in Cytb₆f complex assembly by mediating the interaction between PetB and the PetA-containing subcomplexes to form an assembly intermediate that contains all four core subunits (Fig. 9). DEIP1 could act as a scaffolding protein that brings PetA and PetB together, possibly aided by DEIP1 self-dimerization. The exact protein composition of the DEIP1-containing lowly abundant intermediate VI (containing PetA and PETC; Fig. 6a) as well as the composition of the assembly intermediates II and III (containing PetB) remains to be determined in future analyses.

In summary, our work reported here has identified an essential assembly factor for the biogenesis of the Cytb₆f complex in thylakoid membranes. By interacting with the two cytochromes of the complex, cytochrome f (PetA) and cytochrome b₆ (PetB), the DEIP1 protein appears to promote the association of these two key subunits of the core complex and facilitate the assembly of at least two transiently accumulating intermediate complexes in Cytb₆f biogenesis. The identification of DEIP1 will facilitate future investigations into the challenging problem of resolving the assembly pathway of the Cytb₆f complex.

## Methods

**Plant material and growth conditions.** *Arabidopsis thaliana* plants were grown in 0.5× MS medium supplemented with 1% sucrose under long-day conditions (16 h light, 8 h dark) at 22 °C, and with a light intensity of 5 µmol m⁻² s⁻¹ (low light; LL), 95 µmol m⁻² s⁻¹ (moderate light; ML) or 600 µmol m⁻² s⁻¹ (high light; HL). The ecotype Col-0 (CS60000) was used as wild type. The T-DNA insertion mutants *deip1-1* (SALK_037263) and *deip1-2* (SALK_0672267) were obtained from NASC. The transgenic line expressing a chloroplast-targeted GFP under the control of the *UBIQUITIN10* promoter was described previously[65]. All other transgenic lines were generated in this study.

**Phylogenetic tree construction and sequence alignment.** Putative orthologs of AT2G27290 were identified in PhyloGenes[66], and protein sequences were retrieved from UniProt[67]. Phylogenetic tree construction and sequence alignment were conducted using phylogeny.fr[68,69]. Detailed information on the proteins included in the analyses is provided in Supplementary Table 1.

**Etiolation and de-etiolation kinetics.** Arabidopsis seeds were surface sterilized and stratified at 4 °C in the dark for 2–3 days[54]. The plates were placed in the light for 1 h, followed by 7 days in dark. After 7 days, the plants were placed under continuous moderate light (95 µmol m⁻² s⁻¹), and samples were taken after 0, 3, 6, 12, 24, and 48 h of illumination.

**RNA extraction, cDNA synthesis, and transcript analysis by qRT-PCR.** Total RNA was extracted using the NucleoSpin® RNA plus Plant kit (Macherey-Nagel). For cDNA synthesis, 0.5 µg of total RNA was employed and the reverse transcription reaction was conducted using the SuperScriptIII reverse transcriptase and equal molar amounts of both oligo-dT and random hexamer primers. The primers used in this study for DNA amplification are listed in Supplementary Table 1. Transcript accumulation in qRT-PCR was calculated by the ΔΔCt method[70], using the accumulation of *UBQ10* as internal control.

**Genome editing with CRISPR-Cas9.** To generate *deip1* deletion mutants, we employed the CRIPR-Cas9 method as previously described[71]. Briefly, the single-guide RNAs (sgRNAs) were incorporated in the sequences of primers cc_AT2G27290_sgRNA1_For and cc_AT2G27290_sgRNA1_(Rev_Comp) (Supplementary Table 1), and used for PCR amplification using as a template the previously described plasmid pHEE2E-TRI[71]. The resulting PCR product was digested with BsaI, and ligated into the BsaI digested binary vector pJF1031[71]. Col-0 plants were transformed by Agrobacterium-mediated transformation[72], and selected in the presence of 20 µg mL⁻¹ hygromycin. The deletion of the targeted sequence was confirmed by genomic DNA extraction, PCR amplification with primers cl_AT2G27290_For and cl_AT2G27290_Rev (Supplementary Table 2) and DNA sequencing.

**Generation of P$_{UBQ10}$::DEIP1-GFP/deip1-1 complemented lines**. The GreenGate cloning method was employed to generate the complemented lines[73]. To this end, the coding sequence of *DEIP1* was amplified by PCR with primers cl_AT2G27290_For and cl_AT2G27290_Rev. The purified amplification product and the vector pGGC000 were mixed in 5:1 ratio, digested with BsaI and ligated with T4 DNA ligase. The resulting plasmid pGGC-DEIP1 was mixed with the vectors pGGA006, pGGB003, pGGD001, pGGE001, pGGF005, and pGGZ001 in equimolar ratios, digested with BsaI, and ligated with T4 ligase. The *deip1-1* heterozygous mutants were transformed with the resulting plasmid pGGZ-P$_{UBQ10}$::DEIP1-GFP by Agrobacterium-mediated transformation and transgenic lines were selected in the presence of 20 μg mL$^{-1}$ hygromycin. Isolation of P$_{UBQ10}$::DEIP1-GFP/*deip1-1* lines (*deip1*-C lines) was carried out using the segregation of the resistant cassette for pGGZ- P$_{UBQ10}$::DEIP1-GFP, and PCR-based genotyping with the primers cl_AT2G27290_For and gt_AT2G27290-Intron_Rev for the wild-type allele, and cl_AT2G27290_For and LB1.3 for the T-DNA mutant allele (*deip1-1*) (Supplementary Table 2).

**Confocal laser scanning microscopy (CLSM)**. Confocal microscopy was conducted with a Leica TCS SP8 microscope. The excitation/emission settings used were 488/505-535 nm for GFP, 512/520-540 nm for YFP 566/50-600 nm for RFP, and 488/680-720 nm for chlorophyll fluorescence. All images were acquired using sequential scanning settings. For analysis of chloroplast ultrastructure, chlorophyll autofluorescence was employed, and 5–10 z-stacks of 0.2–0.5 μm were acquired at low resolution, to avoid quenching of the fluorescence. The images were combined using the ImageJ software, considering the maximum intensity of each stack.

**Protein extraction and immunoblotting**. Plant material from 7-day-old seedlings or 5-week-old plants was ground in liquid nitrogen, and total protein extracts were obtained by adding TKMES buffer [100 mM Tricine-KOH pH 7.5, 10 mM KCl, 1 mM MgCl$_2$, 1 mM EDTA, 10% (w/v) sucrose, 0.2% (v/v) Triton X-100, 1 mM DTT, 2× protease inhibitor cocktail (Roche)][13]. For immunoblot analyses, total protein extracts or purified thylakoid proteins were resolved in 12.5% SDS-PAA gels and blotted onto PVDF membranes. All antibodies used in this study are listed in Supplementary Table 3.

**Thylakoid isolation and fractionation**. Plant material (leaves or cotyledons) was harvested and ground at 4 °C under dim light in homogenization buffer [50 mM HEPES pH 7.5, 330 mM sorbitol, 1 mM MgCl$_2$, 2 mM EDTA, 5 mM sodium ascorbate and 0.1% (w/v) BSA]. The homogenized tissue was filtered through two Miracloth layers, and centrifuged at $6000 \times g$ for 4 min at 4 °C. The pellet was resuspended in shock buffer [50 mM HEPES pH 7.5, 5 mM sorbitol and 5 mM MgCl$_2$], and incubated on ice in darkness for 5 min. The samples were then centrifuged at $6000 \times g$ for 4 min at 4 °C, the resulting pellet was washed and resuspended in storage buffer [50 mM HEPES pH 7.5, 100 mM sorbitol, and 10 mM MgCl$_2$][74].

For thylakoid subfractionation, thylakoid samples equal to 1 mg of chlorophyll were resuspended in storage buffer to a final concentration of 0.5 mg mL$^{-1}$. The thylakoid samples were mixed slowly with an equal volume of 2% (w/v) digitonin and incubated on ice with stirring for 5 min. The suspension was then homogenized in a potter and centrifuged at $1000 \times g$ for 2 min at 4 °C to remove insoluble material. Subsequently, the soluble fraction was centrifuged at $40,000 \times g$ for 30 min at 4 °C, the pellet was resuspended in storage buffer (grana core fraction), and the soluble fraction was transferred to a new tube and centrifuged at $150,000 \times g$ for 90 min at 4 °C in a fixed-angle rotor. The loose pellet (grana margins), and the strongly anchored pellet (stroma lamellae) were diluted in storage buffer[75].

**Salt washes to assess protein anchoring in membranes**. Thylakoid samples equivalent to 0.5 mg mL$^{-1}$ of chlorophyll were incubated with 1 M NaCl for 1 h at 4 °C. The thylakoid samples were then centrifuged at $12,000 \times g$ for 5 min, and proteins present in pellets and soluble fractions were resolved by electrophoresis in a 12.5% SDS-PAA gel[76].

**Trypsin digestion for protein topology determination**. For trypsin digestion, thylakoid samples equivalent to 0.5 μg chlorophyll μL$^{-1}$ were incubated with or without 10 μg mL$^{-1}$ trypsin at 37 °C for 15 min. Sample buffer for electrophoresis was added and the samples were immediately boiled at 98 °C for 5 min. Finally, the samples were centrifuged for 5 min at $16,000 \times g$ to remove insoluble material, and proteins were resolved in a 12.5% SDS-PAA gel[76].

**BN-PAGE and 2D-SDS-PAGE**. Thylakoid membranes equivalent to 1 μg chlorophyll μL$^{-1}$ were resuspended in 25BTH20G [25 mM BisTris/HCl (pH 7.0), 20% (w/v) glycerol and 2× cOmplete EDTA free protease inhibitor (Roche)] and solubilized with an equal volume of 2% (w/v) n-dodecyl-β-maltoside (β-DDM) for 2 min at 4 °C in the dark. The solubilized samples were centrifuged at $15,000 \times g$ for 15 min at 4 °C, and the soluble fraction was transferred and mixed in a 1:10 volume ratio with Serva Blue G buffer [100 mM BisTris/HCl (pH 7.0), 0.5 M epsilon-aminocaproic acid (ACA), 30% (w/v)

sucrose and 50 mg mL$^{-1}$ Serva Blue G][74]. The protein complexes were resolved in 5–12.5% or 8–13.9% gradient gels, as stated. For the second dimension, the strips were solubilized in Laemmli buffer [138 mM Tris-HCL pH 6.8, 6 M urea, 22.2% glycerol, 4.3% w/v SDS and 100 mM DTT] for 1.5 h and resolved by 15% SDS-PAGE. For silver staining, gels were washed twice with deionized water, and fixed overnight with a solution containing 40% (v/v) methanol and 10% (v/v) acetic acid. The gels were rinsed three times for 20 min with 30% ethanol and incubated with 0.02% (w/v) sodium thiosulfate for 1.5 min. The gels were washed 3 times with deionized water for 20 s and then incubated in the dark with silver nitrate solution [0.2% (w/v) silver nitrate and 0.02% (w/v) formaldehyde] for 20 min. Finally, the gel was washed 3 times with deionized water for 20 s and incubated with developer solution (3% (w/v) sodium carbonate anhydrous, 0.05% (w/v) formaldehyde and 0.0005% (w/v) sodium thiosulfate]. The developer solution was neutralized with 0.05% (w/v) glycine once the spots appeared in the gel[74]. Immunodetection was conducted as described above.

**Measurement of chlorophyll contents**. Chlorophyll contents were determined spectrophotometrically[77]. Briefly, plant materials were ground in liquid nitrogen, and the powder was weighed. Chlorophylls were extracted with 80% (v/v) acetone at 4 °C for 1 h in the dark. The samples were then centrifuged and the absorbance of the supernatant was measured.

**Transmission electron microscopy (TEM)**. Cotyledons from Col-0 and *deip1* mutants were fixed in Karnovsky's solution [2.5% (w/v) glutaraldehyde, 2% (w/v) paraformaldehyde, 0.1 M sodium cacodylate buffer pH 7.4]. The samples were post-fixed with 2% (w/v) osmium tetroxide for 2 h on ice, washed for 15 min with 0.1 M sodium cacodylate buffer pH 7.4, and washed trice with deionized water for 10 min each. The samples were then post-stained with 2% (w/v) uranyl acetate for 2 h at 4 °C, washed for 10 min with miliQ water and dehydrated with increasing concentrations of ethanol [10 min 30% (v/v); 15 min 50%; 30 min 70%, 40 min 80%; 40 min 90% and overnight 100%]. The dehydrated samples were infiltrated with Agar100 epoxy resin (agar scientific) and embedded by incubation at 60 °C for 48 h. Ultrathin sections (50–70 nm) were cut with diamond knives, contrasted with 2% (w/v) uranyl acetate and lead citrate, and examined in a Zeiss EM 912 Omega transmission electron microscope (Carl Zeiss, Oberkochen, Germany)[78].

**Imaging pulse amplitude modulation (imaging PAM) measurements**. For chlorophyll *a* fluorescence analyses, seedlings were dark-adapted for 15 min prior to PAM measurements, using the IMAGING-PAM M-Series (Heinz Walz GmbH, Effeltrich, Germany). The following settings were used for data collection: Gain = 1, Set Damping = 2, Measure Light = 2, Means Freq = 1. The following program was used for measurement of plants: F$_v$/F$_m$ estimation, pause for 5 s, actinic light set at 56 PAR for 10 min and measurement (low light; LL), followed by actinic light set at 461 PAR for 10 min and a final measurement (high light; HL).

**Ribosome profiling**. Four hundred milligrams of frozen plant tissue was homogenized in liquid nitrogen and thawed in 4 mL lysis buffer [0.2 M sucrose, 0.2 M KCl, 40 mM Tris-OAc pH 8.0, 10 mM MgCl$_2$, 10 mM 2-mercaptoethanol, 2% (v/v) polyoxyethylene(10)-tridecyl ether, 1 % (v/v) Triton X-100, 100 μg mL$^{-1}$ chloramphenicol, 100 μg mL$^{-1}$ cycloheximide]. A 0.5 mL aliquot of the lysate was used for total RNA isolation and the rest was incubated at 20 °C for 1 h with 600 U of micrococcal nuclease (MNase). Subsequently, the lysate was layered onto a 1 mL sucrose cushion [30% (w/v) sucrose, 0.1 M KCl, 40 mM Tris-acetate, pH 8.0, 15 mM MgCl$_2$, 5 mM 2-mercaptoethanol, 100 μg mL$^{-1}$ chloramphenicol, and 100 μg mL$^{-1}$ cycloheximide], and centrifuged for 1.5 h at 50,000 rpm and 4 °C in an SW55 Ti rotor to pellet monosomes. RNA was isolated from the monosome pellet and the 0.5 mL aliquot using Trizol (Fisher Scientific) according to the manufacturer's instructions. Subsequently, RNAs from the monosome pellet ranging from 23 nt to 45 nt (including ribosome footprints) were size selected in a 12% denaturing PAA gel (19:1, acrylamide:bisacrylamide). Total RNA was chemically fragmented by incubation in fragmentation buffer [40 mM Tris-OAc, pH 8.3, 100 mM KOAc, and 30 mM Mg(OAc)$_2$] for 12.5 min at 85 °C. 4 μg purified ribosome footprints and 3.5 μg fragmented total RNA derived from mutant and wild-type plants were labeled with Cy3 and Cy5 (ULS Small RNA Labeling Kit, Kreatech Diagnostics), respectively, according to the manufacturer's instructions and hybridized to a custom tiling microarray that covers all reading frames in the Arabidopsis chloroplast genome with overlapping 50-nt oligonucleotide probes[79]. Microarrays were scanned with a GenePix 4000B microarray scanner.

Data analysis was performed as follows: For each dataset, the average of the local background-subtracted single-channel signals (F635-B635 and F532-B532, respectively) was normalized to a constant value (4000) to remove biases introduced by technical variation. The average of the probe signals for each reading frame was calculated and log$_2$-transformed. The relative abundances of ribosome footprints and total RNA were calculated for each reading frame by normalization of the average of each reading frame to the average signal of all reading frames. Translation efficiencies were calculated for each reading frame by subtracting the summarized log$_2$-transformed signals of the total RNA from the summarized log$_2$-transformed signals of ribosome footprints. The average and standard deviation of

relative abundances of ribosome footprints, total RNA, and translation efficiency were calculated for each reading frame from three biological replicates. Significances of changes were calculated using the Empirical Bayes method in the LIMMA package[80] and the *P* values were adjusted for multiple testing according to FDR approach to correct for multiple comparisons[81]. Genes with twofold or more than twofold change and an adjusted *P* value < 0.05 were considered significantly differentially expressed.

**BS³ crosslinking**. For BS³ crosslinking experiments, 7-day-old seedlings grown under standard conditions were used as starting material. Membrane fractions equivalent to 0.5 µg chlorophyll µL⁻¹ obtained from complemented line *deip1-C4*, or soluble fractions equivalent to 5 µg of protein µL⁻¹ obtained from ctpGFP, were incubated with 1.5 mM BS³ for 1 h on ice. The samples were then resuspended in reducing sample buffer for PAA gel electrophoresis, boiled at 98 °C for 5 min, and separated in a 10% PAA gel.

**DSP crosslinking and co-immunoprecipitation (co-IP)**. Co-immunoprecipitation experiments were performed using the GFP-Trap Kit (Chromotek). Thylakoids extracted from the wild type and the *deip1-C4* line were treated with 1.5 mM DSP for 30 min at room temperature, solubilized in RIPA buffer and centrifuged to remove insoluble material. The supernatant was mix with the GFP magnetic beads and incubated in rotation shaker at 4 °C in the dark for 12 h. The magnetic beads were then washed five times with RIPA buffer, and the bound proteins were eluted by incubation with sample buffer for PAA gel electrophoresis containing 100 mM DTT, and boiled at 95 °C for 10 min. The input and the final elution from the co-IPs were resolved in 12.5% SDS-PAA gels.

**Bimolecular fluorescence complementation (BiFC) assays**. For BIFC analysis[82], the coding sequences of *DEIP1*, *petA*, *petB*, *PETC*, and *petD* (without the stop codon) were amplified with the primers listed in Supplementary Table 1. For *DEIP1*, the coding region including its native transit peptide sequence was amplified, whereas the sequence for the pea RBCS transit peptide was introduced in front of the coding regions of the Cyt*b₆f* subunits. Plasmid vectors for BiFC analyses were generated by LR clonase reactions using the vector pBIFCt-2in1-CC as destination vector[82]. Five-week-old tobacco plants (*N. benthamiana*) were transformed by co-infiltration of *Agrobacterium tumefaciens* strain GV3101 harboring the destination vector for BiFC and strain GV3101 harboring the p19 suppressor of gene silencing[83]. Three days after infiltration, protoplasts were isolated[84] and the fluorescences of YFP and RFP were analyzed by confocal laser-scanning microscopy with the settings described above.

**Reporting summary**. Further information on research design is available in the Nature Research Reporting Summary linked to this article.

## Data availability

The data supporting the findings of this study are available within the paper and its supplementary information files. Genetic material and all raw data are available upon request from the corresponding author. Source data are provided with this paper.

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

## Acknowledgements

We thank Dr. Stephan Greiner (Max Planck Institute of Molecular Plant Physiology) for providing a PetD antibody. We are grateful to Anja Froehlich for excellent technical assistance with TEM experiments. O.S.-I. thanks Dr. Mark Aurel Schöttler for advice on the measurement of photosynthetic parameters. This work was supported by the Max Planck Society and a grant from the Deutsche Forschungsgemeinschaft (FOR 2092; BO 1482/17-2) to R.B. R.Z. is supported by the Max Planck Society and the Deutsche Forschungsgemeinschaft (SFB-TRR175 (A04), ZO 302/5-1).

## Author contributions
O.S-I. and R.B. designed research, O.S-I., D.R., R.G., A.P.H., T.A-M., A.S., and R.Z. performed research. All authors contributed to data analysis. O.S-I., D.R., R.G., A.P.H., T.A-M., R.Z., and R.B. wrote the manuscript.

## Funding

## Competing interests
The authors declare no competing interests.

## Additional information



