## [Peer Review File · Nature Communications]

De-etiolation-induced protein 1 (DEIP1) mediates assembly of the cytochrome b6f complex in ArabidopsisREVIEWER COMMENTS

Reviewer #1 (Remarks to the Author):

Summary, Noteworthy Results, and Significance to the Field:

Photosynthetic complexes in the chloroplast thylakoid membranes are essential for photosynthesis. Although the structural compositions of these photosynthetic complexes have been well recognized, relatively little is known about the assembly of these complexes. This is especially true for the cytochrome b6f complex (Cytb6f), which connects electron transport between photosystem II and photosystem I and couples photosynthetic electron transport with the formation of transmembrane proton gradient.

This manuscript reports the identification of a protein factor (de-etiolation-induced protein 1 [DEIP1]), which is essential for Cytb6f assembly. The *deip1* knock-out mutants show a loss of Cytb6f accumulation and defective Cytb6f assembly. This results in the pigment-deficient phenotype and incapability of photoautotrophic growth under moderate light. These mutant phenotypes could be rescued by introducing the PUBQ10::DEIP1-GFP construct into the mutant plants. To dissect the functions of DEIP1, the authors performed array-based RNA and ribosome profiling and discovered that DEIP1 does not act at the level of Cytb6f subunit expression. Lincomycin infiltration and subsequent SDS-PAGE and immunoblot analysis showed that DEIP1 does act in the degradation or stability of Cytb6f subunits either.

Interestingly, blue native-polyacrylamide gel electrophoresis (BN-PAGE) showed that DEIP1 may interact with Cytb6f core subunits PetA and PetB and act as an assembly factor for Cytb6f. The interactions of DEIP1 with PetA and PetB were confirmed by bimolecular fluorescence complementation (BiFC) assays and crosslinking co-immunoprecipitation (co-IP). Based on all these complementary data, the authors conclude that DEIP1 mediates Cytb6f assembly in Arabidopsis. This manuscript provides an entry point into the study of the assembly pathway of Cytb6f, a protein complex essential to photosynthetic electron transport, which is significant to the field.

The reviewers have the following major and minor comments.

Major comments:

The Abstract could be revised to include additional results from the manuscript. The current abstract is too brief.

Page 5, line 148-150. Consider including (1) a phylogenetic tree of DEIP1 proteins from different embryophytes; (2) alignment of a few representative DEIP1 proteins and the putative orthologue in *Chlamydomonas*, to support this statement.

Figure 1G. The protein band below DEIP1-GFP in the trypsin-treated sample should be labeled. According to Figure 1G, the amount of the thylakoid membrane protein PsaB detected in the trypsin-treated sample is much lower than that detected in the non-treated sample. The authors are recommended to explain why PsaB, a thylakoid membrane protein, is not protected from trypsin digestion. This would help readers understand the proposed topology of DEIP1-GFP.

Page 15, Figure 1 legend, line 433. Page 37, Supplementary Figure S2 legend, line 978. Page 38, Supplementary Figure S3 legend, line 999. Page 39, Supplementary Figure S4 legend, line 1008-1009 and 1015. Page 40, Supplementary Figure S5 legend, line 1033-1034. Page 41, Supplementary Figure S6 legend, line 1045. The authors stated that both ANOVA test and Tukey post-test were performed for Figure 1B, and Supplementary Figures S2A, S3B, S4B, S4D, S5A, S5B, S5C, S5D, S6A, S6B, S6C, and S6D. However, only one set of asterisks is shown in these figure panels. Not sure whether this set of asterisks (in each of these figure panels) is based on ANOVA test or Tukey post-test. The authors

are recommended to double check their lab notebooks and excel files. If only one statistical test (e.g., Tukey post-test) was performed, please delete the other test (e.g., ANOVA test) from each figure legend.

Page 20, Figure 4. The BN-PAGE gels in Figure 4A and 4B were apparently subjected to different durations of electrophoresis. The duration of the BN-PAGE gel in Figure 4A seemed to be a lot longer than that in Figure 4B, which resulted in the absence of PSII megacomplexes or PSI-NDH in the BN-PAGE gel in Figure B. The authors are strongly recommended to justify such differences.

Page 22, Figure 5C. Page 46, Supplementary Figure S11. Some of the immunoblots (e.g., PetA, PetB, and PetD) in these two figures are of poor quality. It is hard to tell whether the bands in the IP lanes are the results of actual interactions or carryovers from overloaded neighboring input lanes. The presence of PetA, PetB, and PetD bands in the wild-type IP lanes (which should be absent) suggests that carryovers from neighboring input lanes are problematic.

Page 37, Supplementary Figure S2 legend, line 974. The authors are recommended to double check the order of DEIP1, LPOR, and LHCB2 expression levels in Supplementary Figure S2A. If they are in the order of DEIP1, LPOR, and LHCB2, the authors should change "DEIP1, LHCB2 and LPOR" on line 974 to "DEIP1, LPOR and LHCB2".

Page 37, Supplementary Figure S2 legend, line 974-987. Furthermore, the control groups used in ANOVA and/or Tukey post-test in Supplementary Figure S2A are not clear. Please include such information in the figure legend. If mRNA levels of DEIP1 (black), LPOR (grey) and LHCB2 (clear) at 0 hour after etiolation serve as the control groups in these two statistical tests, why the clear bars (LHCB2) at 6 and 12 h of de-etiolation do not have asterisks above them? These two bars appear significantly taller than the clear bar (LHCB2) at 0 hour after etiolation.

Page 37, Supplementary Figure S2 legend, line 977. The authors stated that "The (qRT-PCR) data were normalized to the two housekeeping genes UBQ10 and EF1a." It's technically impossible to normalize qRT-PCR data to two different housekeeping genes. The authors are recommended to double check their lab notebooks and revise this statement accordingly. (It is achievable to normalize qRT-PCR data to one housekeeping gene, either UBQ10 or EF1alpha, not both.)

Page 40, Supplementary Figure S5. The control groups used in ANOVA and/or Tukey post-test in Supplementary Figure S5A, S5B, S5C, and S5D are not clear. Please include such information in the figure legend. For example, in Supplementary Figure S5B, are the three asterisks above the light grey bar (i.e., NPQ value of deip1-1 under high light) based on the statistical comparison with wild type NPQ value under high light or deip1-1 NPQ value in the dark? Similarly, in Supplementary Figure S5C, are the three asterisks above the light grey bar (i.e., qL value of deip1-1 under low light) based on the statistical comparison with wild type qL value under low light or deip1-1 qL value in the dark?

Page 41, Supplementary Figure S6. The control groups used in ANOVA and/or Tukey post-test in Supplementary Figure S6A, S6B, S6C, and S6D are not clear. Please include such information in the figure legend.

Page 42, Supplementary Figure S7 legend, line 1054-1062. Page 43, Supplementary Figure S8 legend, line 1071-1083. The authors are recommended to state whether they performed Tukey post-test on the chlorophyll data in Supplementary Figures S7B and S8A and whether any of the comparisons are statistically significant, in the figure legend.

Minor comments:

Page 4, line 126. What mature complexes? Please be clear.

Page 4, line 129. Consider inserting "The" before "deip1 mutants" to avoid beginning the sentence

with a lower-case word.

Page 8, line 275 and other places. The authors are recommended to change to "7 day-old" to "7-day-old" throughout the manuscript.

Page 9, line 282. Authors are recommended to spell out PAA (polyacrylamide) at first mention, to minimize confusion.

Page 9, line 309-312. Authors are recommended to cite the figure number(s) to support this statement. Should it be Figure 4?

Page 10, line 316-317. How long was the lincomycin treatment?

Page 10, line 321. Consider changing "the Cytb6f complex" to "Cytb6f subunits".

Page 10, line 341-342. How do the authors know that DEIP1 is present as a monomer according to Figure 5B? By size?

Page 11, line 370-375. Consider revising and splitting this sentence to avoid using a lengthy sentence within the parentheses.

Page 11, line 376-378. This sentence is confusing. I think "inform" on line 376 should be "in". Please revise accordingly.

Page 11, line 380-382. This statement (grana stacking is mainly mediated by the accumulation of LHCs) is not well supported by data in this manuscript or the cited reference (49). Please revise.

Page 12, Line 401-404. This sentence is confusing. Should "PetB homodimer" on line 402 be "PetA/PetB heterodimer" actually? If not, this sentence still needs to be revised to minimize such confusion.

Page 12, Line 406-408. According to this sentence, PetA is assembled before PetB. The authors may want to make this order very clear in the manuscript.

Page 408-410. This sentence is confusing. It is not clear to readers which event(s) happen first and which event(s) happen next. BTW, what does "its" on line 410 refer to? Please revise.

Page 15, Figure 1 legend, line 447. The experimental procedure of salt wash of thylakoid samples was not described in the Materials and Methods section.

Page 18, Figure 3. The authors may want to explain why they performed SDS-PAGE and immunoblot analysis with both total protein preparation (Figure 3B) and thylakoid membrane protein preparation (Figure 3C).

Page 20, Figure 4. The authors are recommended to list the full name of dPSII (dimeric PSII?) in the figure legend. There are two typos in Figure 4A and 4B. "LCHII(a)" should be "LCHII(a)" in both places.

Page 22, Figure 5. The deip1-C4 plants express the PUBQ10::DEIP1-GFP construct. The authors used anti-GFP antibody to detect the recombinant DEIP1-GFP protein in deip1-C4 in Figure 5B. Therefore, the authors are recommended to replace the label "DEIP1" with "DEIP1-GFP" and replace the label "hmwDEIP1" to "hmwDEIP1-GFP" in this figure panel.

Page 25, Line 568, 569, 579. The authors are recommended to use standard units for light intensity in these three and other places (if exist), i.e., replace "uE" with "umol photons".

Page 25, Line 592. The authors are recommended to spell out sgRNAs at first mention: single guide RNAs.

Page 26, Line 597. The authors are recommended to insert a space between "transformation" and reference number "61".

Page 26, Line 597 and 610. The authors used "ug ml⁻¹" for the hygromycin concentration on line 597 and "ug/ml" for the hygromycin concentration on line 610. Please be consistent.

Page 27, Line 636. The authors are recommended to insert a space between "procedures" and reference number 63.

Page 27, line 640, 642, 648, 650, and 651. Page 28, line 666. The authors are recommended to replace "40.000g" with "40,000g" and replace "150.000g" with "150,000g". For consistency, the authors are also recommended to replace "6000g" with "6,000g" and replace "15000g" with "15,000g".

Page 28, Line 664. The authors are recommended to insert a space between "protocols" and reference number 63.

Page 28, Line 673-676. The authors are recommended to indicate whether seedling root is included in the tissue used for chlorophyll extraction. Furthermore, the authors are recommended to delete "7 day-old" from line 673 because authors had extracted chlorophyll from 7-day-old, 12-day-old (Supplemental Figure S7), and 5-week-old (Supplemental Figure S8) plants.

Page 28, Line 679. The authors are recommended to change "Cotyledon" to "Cotyledons".

Page 28, Line 693. The authors are recommended to remove "7 day-old" from line 693, as the authors had performed chlorophyll a fluorescence analysis with both 7-day-old (Supplemental Figure S5?) and 12-day-old (Supplemental Figure S7) seedlings.

Page 29, line 708. The number 3 is BS3 should be superscripted.

Page 30, Line 731. The authors are recommended to change "5" to "Five" to avoid beginning a sentence with a numeral.

Page 38, Supplementary Figure S3B. Page 39, Supplementary Figure S4D. It is not clear to readers whether the single set of asterisks in Supplementary Figures S3B and S4D is for chlorophyll a, chlorophyll b, total chlorophyll, or chlorophyll a:b ratio. The authors are recommended to include such information in the figure legend.

Page 38. Supplementary Figure S3 legend, line 992-999. The authors are recommended to include the age of the seedlings in the figure legend.

Page 40. Supplementary Figure S5 legend, line 1029-1035. The authors are recommended to indicate the age of the seedlings (7-day-old?) used for chlorophyll fluorescence measurements, in this figure legend. (Apparently, authors had also performed chlorophyll fluorescence measurements with 12-day-old seedlings (Supplementary Figure S7)).

Page 44. Supplementary Figure S9. The authors are recommended to label PSII, Cytb6f, PSII, LHCII

complexes next to the four BN-PAGE gel strips in this figure to help readers understand.

Reviewer #2 (Remarks to the Author):

The DEIP1 gene was discovered by its early induction during the de-etiolation process. The knock-out mutants showed a specific loss of the cytochrome b6f complex in Arabidopsis. The DEIP1 protein interacted with PetA and PetB. The authors proposed a model, in which the DEIP1 homodimer recruits the assembly intermediate including PetA to the PetB homodimer.

It is not easy to determine the function of factor, which is necessary for the assembly of the thylakoid membrane complex. The authors of this manuscript performed an excellent work, which is possible in the study using land plants. I list only some minor points to improve the manuscript further.

- 1) Is it possible to detect the accumulation of the assembly intermediates including PetA and PetC ... (complex IV) in the KO mutant?
- 2) Ls228-235. The b6f complex restricts the rate of electron transport by monitoring the luminal acidification (photosynthetic control). It is reasonable that the photosynthetic control is enhanced when the level of the b6f complex is low and too strong photosynthetic control damages PSII as observed in the Arabidopsis pgr1 mutant.
- 3) Figure 4C and D. This is an excellent work. It is clear that DEIP1 is not involved in the chloroplast gene expression.
- 4) Ls400-401. How is complex V?
- 5) L230. Supplementary Figure 5C.
- 6) L254. Is it appropriate to use "in vitro culture" for the heterotrophic culture?
- 7) An illustration for the assembly model may be helpful for readers.

Reviewer #3 (Remarks to the Author):

The manuscript "De-etiolation-induced protein 1 is an essential assembly factor of cytochrome b6f in Arabidopsis" by Sandoval-Ibáñez et al. describes the assignment of DEIP1 and its characterization in the cytochrome b6f complex assembly. This protein factor was found by screening highly time-resolved RNAseq datasets of greening tobacco leaves for genes whose transcripts responded early to the de-etiolation signal. The corresponding putative gene product is conserved in embryophyte (AT2G27209 in Arabidopsis) and the green alga *Chlamydomonas reinhardtii*. Thus, to address the physiological function of the DEIP1 gene product, the authors characterized the mutants deficient in DEIP1 obtained from the collections of T-DNA insertion lines and by a genome-edited line harboring a large deletion of 1084 bp in the coding region of AT2G27209 in Arabidopsis. The localization in the chloroplasts and characterization of the mutants deficient in DEIP-1 are reported. Since PetA, PetB, and PetD are co-immunoprecipitated with DEIP-1, the authors concluded that DEIP-1 is essential for Cytb6f complex assembly. The authors did experiments carefully and wrote the manuscript logically. The presented data suggest that this factor is required for the complex assembly rather than transcription and translation of Cytb6f subunits. This reviewer evaluates that the conclusion is supported by the experimental results presented in this manuscript. This assembly factor is new because this is the first report on the assembly factor of Cytb6f subunits. However, the authors did not assign that DEIP1 is directly involved in the assembly without showing that PetA, PetB, and PetD pull-downed with DEIP1 are newly synthesized.

Minor points.

1. Is DEIP1 conserved in cyanobacteria?
2. L209; does this mean that DEIP1 has one or more transmembrane helix? A hydropathy plot should

be shown.

3. It is important to show the parameters of PAM measurements. However, these are affected indirectly by the mutation. I wonder why the authors did not show light-induced fluorescence induction kinetics, which easily provides us more direct evidence of whether the mutants have PSI/Cytb6f or PSII activity.

4. Fig. 5A shows the separation of Cytb6f subcomplexes (I,-VI). This reviewer thinks that it cannot be excluded that these bands are derived from disassembled subcomplexes rather than assembly intermediates. It is necessary to describe more details of the band I-VI. The positions of size marker proteins should be indicated to know the apparent sizes of I-VI subcomplexes. It looks strange that VI lacks PetD, but V contains all major subunits. The Cytb6f complex may be a dimer, so is a monomeric band detected?

5. The hmwDEIP1 in Figure 5B can be the dimer, or it is possible that the large complex associates one or some of PetA, PetB, and PetD.

We would like to thank the three reviewers for their constructive criticisms and suggestions. As you will see, we have performed additional experimentation and made appropriate textual revisions to fully address all points raised by the referees.

Below please find a point-by-point response to the reviewers' comments (also uploaded as a separate document).

Reviewers' Comments (with our response in red; changes to the text are highlighted in the marked-up copy of the manuscript in yellow):

Reviewer #1:

Summary, Noteworthy Results, and Significance to the Field:

Photosynthetic complexes in the chloroplast thylakoid membranes are essential for photosynthesis. Although the structural compositions of these photosynthetic complexes have been well recognized, relatively little is known about the assembly of these complexes. This is especially true for the cytochrome b6f complex (Cytb6f), which connects electron transport between photosystem II and photosystem I and couples photosynthetic electron transport with the formation of transmembrane proton gradient.

This manuscript reports the identification of a protein factor (de-etiolation-induced protein 1 [DEIP1]), which is essential for Cytb6f assembly. The *deip1* knock-out mutants show a loss of Cytb6f accumulation and defective Cytb6f assembly. This results in the pigment-deficient phenotype and incapability of photoautotrophic growth under moderate light. These mutant phenotypes could be rescued by introducing the PUBQ10::DEIP1-GFP construct into the mutant plants. To dissect the functions of DEIP1, the authors performed array-based RNA and ribosome profiling and discovered that DEIP1 does not act at the level of Cytb6f subunit expression. Lincomycin infiltration and subsequent SDS-PAGE and immunoblot analysis showed that DEIP1 does act in the degradation or stability of Cytb6f subunits either.

Interestingly, blue native-polyacrylamide gel electrophoresis (BN-PAGE) showed that DEIP1 may interact with Cytb6f core subunits PetA and PetB and act as an assembly factor for Cytb6f. The interactions of DEIP1 with PetA and PetB were confirmed by bimolecular fluorescence complementation (BiFC) assays and crosslinking co-immunoprecipitation (co-IP). Based on all these complementary data, the authors conclude that DEIP1 mediates Cytb6f assembly in Arabidopsis. This manuscript provides an entry point into the study of the assembly pathway of Cytb6f, a protein complex essential to photosynthetic electron transport, which is significant to the field.

The reviewers have the following major and minor comments.

Major comments:

The Abstract could be revised to include additional results from the manuscript. The current abstract is too brief.

Unfortunately, the Abstract already exceeds the word limit (of 150 words) allowed by Nature Communications, and we, therefore, cannot further extend it.

Page 5, line 148-150. Consider including (1) a phylogenetic tree of DEIP1 proteins from different embryophytes; (2) alignment of a few representative DEIP1 proteins and the putative orthologue in *Chlamydomonas*, to support this statement.

As suggested by the Reviewer, we included both a phylogenetic tree and a sequence alignment in the revised version (Supplementary Figures 2 and 3). The results are also briefly described in the main text (p. 5).

Figure 1G. The protein band below DEIP1-GFP in the trypsin-treated sample should be labeled. According to Figure 1G, the amount of the thylakoid membrane protein PsaB detected in the trypsin-treated sample is much lower than that detected in the non-treated sample. The authors are recommended to explain why PsaB, a thylakoid membrane protein, is not protected from trypsin digestion. This would help readers understand the proposed topology of DEIP1-GFP.

We explained the degradation of PsaB and the resistance of PSBO in the figure legend. Additionally, we labelled the degradation product of DEIP1-GFP as DEIP1-GFP* in Figure 1G.

Page 15, Figure 1 legend, line 433. Page 37, Supplementary Figure S2 legend, line 978. Page 38, Supplementary Figure S3 legend, line 999. Page 39, Supplementary Figure S4 legend, line 1008-1009 and 1015. Page 40, Supplementary Figure S5 legend, line 1033-1034. Page 41, Supplementary Figure S6 legend, line 1045. The authors stated that both ANOVA test and Tukey post-test were performed for Figure 1B, and Supplementary Figures S2A, S3B, S4B, S4D, S5A, S5B, S5C, S5D, S6A, S6B, S6C, and S6D. However, only one set of asterisks is shown in these figure panels. Not sure whether this set of asterisks (in each of these figure panels) is based on ANOVA test or Tukey post-test. The authors are recommended to double check their lab notebooks and excel files. If only one statistical test (e.g., Tukey post-test) was performed, please delete the other test (e.g., ANOVA test) from each figure legend.

We included only the Tukey post-test. The ANOVA test was removed.

Page 20, Figure 4. The BN-PAGE gels in Figure 4A and 4B were apparently subjected to different durations of electrophoresis. The duration of the BN-PAGE gel in Figure 4A seemed to be a lot longer than that in Figure 4B, which resulted in the absence of PSII megacomplexes or PSI-NDH in the BN-PAGE gel in Figure B. The authors are strongly recommended to justify such differences.

The reason for the different migration patterns was the PAA gel gradient employed in the experiments. The gel shown in Fig. 4A was run with a gradient of 6-12.5%, while the gel gradient used in Fig. 4B was 8-13.8% (to better resolved the cytochrome *b₆f* complex, *Cyt_{b₆f}*). That said, to make the gels more comparable, we conducted a new gel electrophoresis for Fig. 4B using a 6-12.5% gel gradient, and replaced panel B (Fig. 5B in the revised manuscript).

Page 22, Figure 5C. Page 46, Supplementary Figure S11. Some of the immunoblots (e.g., PetA, PetB, and PetD) in these two figures are of poor quality. It is hard to tell whether the bands in the IP lanes are the results of actual interactions or carryovers from overloaded neighboring input lanes. The presence of PetA, PetB, and PetD bands in the wild-type IP lanes (which should be absent) suggests that carryovers from neighboring input lanes are problematic.

We repeated the co-immunoprecipitation experiment (Supplementary Fig. 12B) leaving empty lands between all samples (to exclude carryover). The results revealed that the presence of weak signals for PetA and PetB in the wild-type sample is likely due to the use of DSP crosslinking, and not due to carryover from neighbouring input lanes. The gel was added as additional replicate to Suppl. Fig. 12 (panel B) and the results are briefly discussed in the text of the revised manuscript (p. 11).

Page 37, Supplementary Figure S2 legend, line 974. The authors are recommended to double check the order of DEIP1, LPOR, and LHCB2 expression levels in Supplementary Figure S2A. If they are in the order of DEIP1, LPOR, and LHCB2, the authors should change "DEIP1, LHCB2 and LPOR" on line 974 to "DEIP1, LPOR and LHCB2".

Changes as suggested.

Page 37, Supplementary Figure S2 legend, line 974-987. Furthermore, the control groups used in ANOVA and/or Tukey post-test in Supplementary Figure S2A are not clear. Please include such information in the figure legend. If mRNA levels of DEIP1 (black), LPOR (grey) and LHCB2 (clear) at 0 hour after etiolation serve as the control groups in these two statistical tests, why the clear bars (LHCB2) at 6 and 12 h of de-etiolation do not have asterisks above them? These two bars appear significantly taller than the clear bar (LHCB2) at 0 hour after etiolation.

We double-checked the data, and the statistical evaluation. Due to the high expression of LHCB2 after 24 and 48 h (50 and 80 times higher than in etiolated tissue), the changes observed at time points 6 and 12 h are not significant (due to the increase in variance).

Page 37, Supplementary Figure S2 legend, line 977. The authors stated that "The (qRT-PCR) data were normalized to the two housekeeping genes UBQ10 and EF1a." It's technically impossible to normalize qRT-PCR data to two different housekeeping genes. The authors are recommended to double check their lab notebooks and revise this statement accordingly. (It is achievable to normalize qRT-PCR data to one housekeeping gene, either UBQ10 or EF1alpha, not both.)

We thank the Reviewer for spotting this error. The results were normalized to *UBQ10*. This has been corrected in both the figure and the Methods section.

Page 40, Supplementary Figure S5. The control groups used in ANOVA and/or Tukey post-test in Supplementary Figure S5A, S5B, S5C, and S5D are not clear. Please include such information in the figure legend. For example, in Supplementary Figure S5B, are the three asterisks above the light grey bar (i.e., NPQ value of *deip1-1* under high light) based on the statistical comparison with wild type NPQ value under high light or *deip1-1* NPQ value in the dark? Similarly, in Supplementary Figure S5C, are the three asterisks above the light grey bar (i.e., qL value of *deip1-1* under low light) based on the statistical comparison with wild type qL value under low light or *deip1-1* qL value in the dark?

Page 41, Supplementary Figure S6. The control groups used in ANOVA and/or Tukey post-test in Supplementary Figure S6A, S6B, S6C, and S6D are not clear. Please include such information in the figure legend.

Information on the control groups for comparison has been added to the legends of all relevant figures.

Page 42, Supplementary Figure S7 legend, line 1054-1062. Page 43, Supplementary Figure S8 legend, line 1071-1083. The authors are recommended to state whether they performed Tukey post-test on the chlorophyll data in Supplementary Figures S7B and S8A and whether any of the comparisons are statistically significant, in the figure legend.

Information on the statistical test has been added to the legends, and non-significant changes (n.s.) are now indicated on the top of the bars.

Minor comments:

Page 4, line 126. What mature complexes? Please be clear.

To make this clearer, we replaced "mature complex" with "fully assembled Cytb₆f complex".

Page 4, line 129. Consider inserting "The" before "deip1 mutants" to avoid beginning the sentence with a lower-case word.

Done.

Page 8, line 275 and other places. The authors are recommended to change to "7 day-old" to "7-day-old" throughout the manuscript.

Changed as suggested.

Page 9, line 282. Authors are recommended to spell out PAA (polyacrylamide) at first mention, to minimize confusion.

Done.

Page 9, line 309-312. Authors are recommended to cite the figure number(s) to support this statement. Should it be Figure 4?

Done (now Figure 5D).

Page 10, line 316-317. How long was the lincomycin treatment?

This is now explicitly stated in the text (incubated in the dark at 4°C for 16 h; p. 10, l. 326).

Page 10, line 321. Consider changing "the Cytb₆f complex" to "Cytb₆f subunits".

Changed as suggested.

Page 10, line 341-342. How do the authors know that DEIP1 is present as a monomer according to Figure 5B? By size?

The predicted MW of DEIP1-GFP is 52 kDa. Electrophoresis was conducted in an SDS-PAA gel and the loading buffer was supplemented with 100 mM DTT. The efficiency of crosslinking is never 100%, and we therefore, cannot rule out the possibility that a subfraction of free DEIP1 is present as a dimer (as suggested by migration of DEIP1 in our native gels and the dimerization observed in our BiFC assays). We clarify this point in the revised manuscript (p. 11, l. 356), and in the Discussion (p. 13, l. 438). Additionally, we included the molecular weights of the marker bands in the immunoblot (Fig. 7A).

Page 11, line 370-375. Consider revising and splitting this sentence to avoid using a lengthy sentence within the parentheses.

Done.

Page 11, line 376-378. This sentence is confusing. I think "inform" on line 376 should be "in". Please revise accordingly.

Corrected.

Page 11, line 380-382. This statement (grana stacking is mainly mediated by the accumulation of LHCs) is not well supported by data in this manuscript or the cited reference (49). Please revise.

We replaced the reference by a more suitable one (ref.52) that better supports the statement.

-Page 12, Line 401-404. This sentence is confusing. Should "PetB homodimer" on line 402 be "PetA/PetB heterodimer" actually? If not, this sentence still needs to be revised to minimize such confusion.

-Page 12, Line 406-408. According to this sentence, PetA is assembled before PetB. The authors may want to make this order very clear in the manuscript.

-Page 408-410. This sentence is confusing. It is not clear to readers which event(s) happen first and which event(s) happen next. BTW, what does "its" on line 410 refer to? Please revise.

We modified the text (and the Discussion section) to better describe the steps involved in *Cytb₆f* complex assembly (p. 13). We also added a schematic working model illustrating the proposed role of DEIP1 in complex assembly (also requested by R#2; Fig. 9).

Page 15, Figure 1 legend, line 447. The experimental procedure of salt wash of thylakoid samples was not described in the Materials and Methods section.

The protocol is now described in the Methods section of the revised manuscript ("Salt washes to assess protein anchoring in membranes", p. 33).

Page 18, Figure 3. The authors may want to explain why they performed SDS-PAGE and immunoblot analysis with both total protein preparation (Figure 3B) and thylakoid membrane protein preparation (Figure 3C).

Done (p. 9, l. 281).

Page 20, Figure 4. The authors are recommended to list the full name of dPSII (dimeric PSII?) in the figure legend. There are two typos in Figure 4A and 4B. "LCHII(a)" should be "LCHII(a)" in both places.

Corrected.

Page 22, Figure 5. The *deip1-C4* plants express the PUBQ10::DEIP1-GFP construct. The authors used anti-GFP antibody to detect the recombinant DEIP1-GFP protein in *deip1-C4* in Figure 5B. Therefore, the authors are recommended to replace the label "DEIP1" with "DEIP1-GFP" and replace the label "hmwDEIP1" to "hmwDEIP1-GFP" in this figure panel.

Changed as suggested.

Page 25, Line 568, 569, 579. The authors are recommended to use standard units for light

intensity in these three and other places (if exist), i.e., replace "uE" with "umol photons".

Done.

Page 25, Line 592. The authors are recommended to spell out sgRNAs at first mention: single guide RNAs.

Done.

Page 26, Line 597. The authors are recommended to insert a space between "transformation" and reference number "61".

Done.

Page 26, Line 597 and 610. The authors used "ug ml⁻¹" for the hygromycin concentration on line 597 and "ug/ml" for the hygromycin concentration on line 610. Please be consistent.

Corrected.

Page 27, Line 636. The authors are recommended to insert a space between "procedures" and reference number 63.

Done.

Page 27, line 640, 642, 648, 650, and 651. Page 28, line 666. The authors are recommended to replace "40.000g" with "40,000g" and replace "150.000g" with "150,000g". For consistency, the authors are also recommended to replace "6000g" with "6,000g" and replace "15000g" with "15,000g".

Corrected.

Page 28, Line 664. The authors are recommended to insert a space between "protocols" and reference number 63.

Done.

Page 28, Line 673-676. The authors are recommended to indicate whether seedling root is included in the tissue used for chlorophyll extraction. Furthermore, the authors are recommended to delete "7 day-old" from line 673 because authors had extracted chlorophyll from 7-day-old, 12-day-old (Supplemental Figure S7), and 5-week-old (Supplemental Figure S8) plants.

Modified as suggested.

Page 28, Line 679. The authors are recommended to change "Cotyledon" to "Cotyledons".

Done.

Page 28, Line 693. The authors are recommended to remove "7 day-old" from line 693, as the authors had performed chlorophyll a fluorescence analysis with both 7-day-old (Supplemental Figure S5?) and 12-day-old (Supplemental Figure S7) seedlings.

Done.

Page 29, line 708. The number 3 is BS3 should be superscripted.

Corrected.

Page 30, Line 731. The authors are recommended to change "5" to "Five" to avoid beginning a sentence with a numeral.

Done.

Page 38, Supplementary Figure S3B. Page 39, Supplementary Figure S4D. It is not clear to readers whether the single set of asterisks in Supplementary Figures S3B and S4D is for chlorophyll a, chlorophyll b, total chlorophyll, or chlorophyll a:b ratio. The authors are recommended to include such information in the figure legend.

This is now specified in the figure legends.

Page 38. Supplementary Figure S3 legend, line 992-999. The authors are recommended to include the age of the seedlings in the figure legend.

Done.

Page 40. Supplementary Figure S5 legend, line 1029-1035. The authors are recommended to indicate the age of the seedlings (7-day-old?) used for chlorophyll fluorescence measurements, in this figure legend. (Apparently, authors had also performed chlorophyll fluorescence measurements with 12-day-old seedlings (Supplementary Figure S7).

Done.

Page 44. Supplementary Figure S9. The authors are recommended to label PSII, Cytb6f, PSII, LHCII complexes next to the four BN-PAGE gel strips in this figure to help readers understand.

Figure revised as suggested.

Reviewer #2:

The DEIP1 gene was discovered by its early induction during the de-etiolation process. The knock-out mutants showed a specific loss of the cytochrome b6f complex in Arabidopsis. The DEIP1 protein interacted with PetA and PetB. The authors proposed a model, in which the DEIP1 homodimer recruits the assembly intermediate including PetA to the PetB homodimer.

It is not easy to determine the function of factor, which is necessary for the assembly of the thylakoid membrane complex. The authors of this manuscript performed an excellent work, which is possible in the study using land plants. I list only some minor points to improve the manuscript further.

1) Is it possible to detect the accumulation of the assembly intermediates including PetA and PetC ... (complex IV) in the KO mutant?

In the revised manuscript, we included a BN-PAGE analysis of samples obtained from the *deip1-1* mutant grown under low-light conditions for 5 weeks (Fig. 6B). The assembly intermediates IV and VI are not detectable in the knockout mutant. Additionally, we observed accumulation of low molecular weight intermediates for PetA and PETC. Finally, we observed a relative decrease in the assembly intermediate III (containing PetB), and accumulation of a novel assembly intermediate for PetD. The data are shown in Fig. 6 and discussed in the revised manuscript (on p. 10, l. 340; p. 13, l. 446).

2) Ls228-235. The b6f complex restricts the rate of electron transport by monitoring the luminal acidification (photosynthetic control). It is reasonable that the photosynthetic control is enhanced when the level of the b6f complex is low and too strong photosynthetic control damages PSII as observed in the Arabidopsis pgr1 mutant.

We thank the Reviewer for this suggestion, which has been incorporated in the Discussion (p. 12, l. 397).

3) Figure 4C and D. This is an excellent work. It is clear that DEIP1 is not involved in the chloroplast gene expression.

We thank the Reviewer for this appreciation of our work.

4) Ls400-401. How is complex V?

We revised the discussion of the assembly pathway (also based on the new data; see point 1), and also included a working model (see point 7). Complex V most likely represents the monomeric *Cytb₆f* containing all core subunits.

5) L230. Supplementary Figure 5C.

Corrected (now Fig. 2A-E).

6) L254. Is it appropriate to use “in vitro culture” for the heterotrophic culture?

Changed to “heterotrophic”, as suggested.

7) An illustration for the assembly model may be helpful for readers.

We included an illustration showing the current working model of *Cytb₆f* complex assembly and the proposed mode of action of DEIP1 (Fig. 9).

Reviewer #3:

The manuscript “De-etiolation-induced protein 1 is an essential assembly factor of cytochrome b6f in Arabidopsis” by Sandoval-Ibáñez et al. describes the assignment of DEIP1 and its characterization in the cytochrome b6f complex assembly. This protein factor was found by screening highly time-resolved RNAseq datasets of greening tobacco leaves for genes whose transcripts responded early to the de-etiolation signal. The corresponding putative gene product is conserved in embryophyte (AT2G27209 in Arabidopsis) and the green alga *Chlamydomonas reinhardtii*. Thus, to address the physiological function of the DEIP1 gene product, the authors characterized the mutants deficient in DEIP1 obtained from the collections of T-DNA insertion lines and by a genome-edited line harboring a large deletion of 1084 bp in the coding region of AT2G27209 in Arabidopsis. The localization in the chloroplasts and characterization of the mutants deficient in DEIP-1 are reported. Since PetA, PetB, and PetD are co-immunoprecipitated with DEIP-1, the authors concluded that DEIP-1 is essential for *Cytb₆f* complex assembly. The authors did experiments carefully and wrote the manuscript logically. The presented data suggest that this factor is required for the complex assembly rather than transcription and translation of *Cytb₆f* subunits. This reviewer evaluates that the conclusion is supported by the experimental results presented in this manuscript. This assembly factor is new because this is the first report on the

assembly factor of Cytb₆f subunits. However, the authors did not assign that DEIP1 is directly involved in the assembly without showing that PetA, PetB, and PetD pull-downed with DEIP1 are newly synthesized.

Minor points.

1. Is DEIP1 conserved in cyanobacteria?

DEIP1 is not found in cyanobacteria. We included a phylogenetic tree and a sequence alignment of DEIP1 and its putative orthologues in embryophyte plants and algae. The figures are included in the supplementary material as Suppl. Figs. S2 and S3, and briefly discussed in the text (p. 5).

2. L209; does this mean that DEIP1 has one or more transmembrane helix? A hydropathy plot should be shown.

We included the hydropathy plot as Suppl. Fig. S1C. DEIP1 contains a single predicted transmembrane segment, also indicated in the sequence alignment (Suppl. Fig. 3).

3. It is important to show the parameters of PAM measurements. However, these are affected indirectly by the mutation. I wonder why the authors did not show light-induced fluorescence induction kinetics, which easily provides us more direct evidence of whether the mutants have PSI/Cytb₆f or PSII activity.

As suggested by the Reviewer, we conducted fluorescence induction kinetics in the wild type, *deip1-1*, *deip1-2* and *deip1-3*. As expected, the fluorescence curves did not differ between the WT and *deip1-2*. By contrast, we observed the typical fluorescence curves characteristic of mutants defective in Cytb₆f complex (e.g., Chaux F et al., 2007; doi:10.1104/pp.17.00421). These results provide additional confirmation of the *deip1* KO mutants being impaired in the Cytb₆f complex. The results are shown in Fig. 2E and discussed in the text (p. 7, l. 228).

4. Fig. 5A shows the separation of Cytb₆f subcomplexes (I,-VI). This reviewer thinks that it cannot be excluded that these bands are derived from disassembled subcomplexes rather than assembly intermediates. It is necessary to describe more details of the band I-VI. The positions of size marker proteins should be indicated to know the apparent sizes of I-VI subcomplexes. It looks strange that VI lacks PetD, but V contains all major subunits. The Cytb₆f complex may be a dimer, so is a monomeric band detected?

Mainly due to fast assembly of the Cytb₆f complex the low turnover of its subunits, there is no firmly established model of complex assembly, and all proposed models (e.g., S. Saif Hasan et al., 2013; doi:10.1016/j.bbabi.2013.03.002) remain speculative to some extent. We also cannot elucidate the exact composition of the putative assembly intermediates observed, and we agree with the Reviewer that some of these complexes represent degradation intermediates (although degradation has been shown to be very slow, with an estimated half-life of more than a week; Hojka, M. et al., Plant Physiol 2014). To shed more light on the identity of the putative intermediates and characterize the role of DEIP1 in their accumulation, we included in the revised manuscript the *deip1-1* mutant in our two-dimensional gel electrophoresis experiments to (Fig. 6B). We observed absence of intermediates IV and VI (containing PetA and PETC), decreased abundance of complex III (containing PetB), and a new complex intermediate containing PetD. Based on these results,

we suggest that DEIP1 contributes to the formation of the monomeric *Cytb_{6f}* complex by a mechanism illustrated in the working model presented in the new Fig. 9 and discussed in more detail on p.13 of the revised manuscript. Also, as requested by the Reviewer, the MW of the marker bands were included in all first-dimension BN-PAGE (Figs. 5 and 6). Finally, the Reviewer is correct in concluding that assembly intermediate V most likely corresponds to the monomeric *Cytb_{6f}* complex containing all core subunits, while the exact composition of intermediate VI currently remains unclear.

5. The hmwDEIP1 in Figure 5B can be the dimer, or it is possible that the large complex associates one or some of PetA, PetB, and PetD.

We have added the MW of the maker bands to allow for better evaluation of the band sizes. We agree with the Reviewer that the high molecular weight complexes (shown in what is now Fig. 7A) may comprise the DEIP1 dimer and DEIP1-containing assembly intermediates. Our 2D-SDS-PAGE analyses (shown in Fig. 6) and our protein-protein interaction analyses (Fig. 8) certainly suggest that the HMW complexes in Fig. 7A represent a heterogeneous population of dimers and putative assembly intermediates (as also suggested by the presence of at least two distinct HMW bands in Fig. 7A).

REVIEWERS' COMMENTS

Reviewer #1 (Remarks to the Author):

The authors have sufficiently addressed most of my comments. Here are some minor comments/corrections:

Page 12, Line 399. Change "the prg1 mutant" to "the pgr1 mutant".

Page 13, Line 425. Change "four cores subunits" to "four core subunits".

Page 17, Line 524-525, Figure 2 legend. The statement "Each measurement was performed with an average of 15 seedlings" contracts with "n = 3". Please revise accordingly.

Page 33, Line 762-763. Consider changing "plant material were" to "plant materials were".

Page 44, Line 1168. Supplementary Figure S2 legend. Consider changing "plants and algae" to "plants and green algae". If I understand this figure correctly, only two algal (green algal) species are included in this figure.

Page 46, Line 1178-1179. Supplementary Figure S3 legend. Consider changing "plants and algae" to "plants and a green alga".

Reviewer #2 (Remarks to the Author):

The authors responded to my comments appropriately.

Reviewer #3 (Remarks to the Author):

The revised manuscript has included the requests by this reviewer properly. One minor correction is required; p13, 425-426; "the four cores subunits PetB and PetD" need to be revised probably by "the four core subunits PetA, PetB, PetC, and PetD". Otherwise, I am satisfied with the revised manuscript.

REVIEWERS' COMMENTS (with our responses in red)

Reviewer #1 (Remarks to the Author):

The authors have sufficiently addressed most of my comments. Here are some minor comments/corrections:

Page 12, Line 399. Change "the prg1 mutant" to "the pgr1 mutant".

Corrected.

Page 13, Line 425. Change "four cores subunits" to "four core subunits".

Corrected.

Page 17, Line 524-525, Figure 2 legend. The statement "Each measurement was performed with an average of 15 seedlings" contracts with "n = 3". Please revise accordingly.

Done (n=3 means three replicates with 15 seedlings each).

Page 33, Line 762-763. Consider changing "plant material were" to "plant materials were".

Changed as suggested.

Page 44, Line 1168. Supplementary Figure S2 legend. Consider changing "plants and algae" to "plants and green algae". If I understand this figure correctly, only two algal (green algal) species are included in this figure.

Changed as suggested.

Page 46, Line 1178-1179. Supplementary Figure S3 legend. Consider changing "plants and algae" to "plants and a green alga".

Changed as suggested.

Reviewer #2 (Remarks to the Author):

The authors responded to my comments appropriately.

Reviewer #3 (Remarks to the Author):

The revised manuscript has included the requests by this reviewer properly. One minor correction is required; p13, 425-426; "the four cores subunits PetB and PetD" need to be revised probably by "the four core subunits PetA, PetB, PetC, and PetD". Otherwise, I am satisfied with the revised manuscript.

Corrected.